# Major advance of South Georgia glaciers during the Antarctic Cold Reversal following extensive sub-Antarctic glaciation

Alastair G.C. Graham[1], Gerhard Kuhn[2], Ove Meisel[2,†], Claus-Dieter Hillenbrand[3], Dominic A. Hodgson[3], Werner Ehrmann[4], Lukas Wacker[5], Paul Wintersteller[6], Christian dos Santos Ferreira[6], Miriam Römer[6], Duanne White[7] & Gerhard Bohrmann[6]

The history of glaciations on Southern Hemisphere sub-polar islands is unclear. Debate surrounds the extent and timing of the last glacial advance and termination on sub-Antarctic South Georgia in particular. Here, using sea-floor geophysical data and marine sediment cores, we resolve the record of glaciation offshore of South Georgia through the transition from the Last Glacial Maximum to Holocene. We show a sea-bed landform imprint of a shelf-wide last glacial advance and progressive deglaciation. Renewed glacier resurgence in the fjords between c. 15,170 and 13,340 yr ago coincided with a period of cooler, wetter climate known as the Antarctic Cold Reversal, revealing a cryospheric response to an Antarctic climate pattern extending into the Atlantic sector of the Southern Ocean. We conclude that the last glaciation of South Georgia was extensive, and the sensitivity of its glaciers to climate variability during the last termination more significant than implied by previous studies.

[1] College of Life and Environmental Sciences, University of Exeter, Exeter EX4 4RJ, UK. [2] Alfred-Wegener-Institut, Helmholtz-Zentrum für Polar- und Meeresforschung, Am Alten Hafen 26, Bremerhaven 27568, Germany. [3] British Antarctic Survey (Natural Environmental Research Council), High Cross, Madingley Road, Cambridge CB3 0ET, UK. [4] Insitute for Geophysics and Geology, University of Leipzig, Talstraße 35, Leipzig 04103, Germany. [5] ETH Zürich, Laboratory of Ion Beam Physics, Schafmattstrasse 20, Zürich CH-8093, Switzerland. [6] MARUM—Center for Marine Environmental Sciences, University of Bremen, Bremen D-28359, Germany. [7] Institute of Applied Ecology, University of Canberra, Canberra, Australian Capital Territory 2601, Australia. † Present address: Department of Earth Sciences, Vrije Universiteit Amsterdam, De Boelelaan 1105, 1081 HV Amsterdam, The Netherlands. Correspondence and requests for materials should be addressed to A.G.C.G. (email: a.graham@exeter.ac.uk).

Since at least the late Miocene ($\sim$9 Ma (refs 1,2)), Antarctica's ice sheets have oscillated between states of relative deglaciation and full glacial conditions on orbital timescales[3]. Although much is known about these patterns of ice-sheet waxing and waning[4,5], the history of glaciations on sub-polar islands is, by comparison, exceptionally poorly understood[6]. The extent and dynamics of ice masses peripheral to the Antarctic ice sheets, although minor in any contributions to sea level change, can provide useful data for optimizing simulations of the Antarctic Peninsula and West Antarctic ice sheets during glacial stages and their sensitivity to atmosphere and ocean warming. In addition to offering up useful boundary conditions, there is an emerging interest in the record of glacial activity on submarine parts of sub-Antarctic islands, which may have served as refuges for marine benthos during past glaciations[7]. There is still considerable debate over the location and duration of terrestrial[8] and marine refugia during Pleistocene glacials in Antarctica, and one hypothesis is that sub-Antarctic islands served as ice-free shelters as well as evolutionary 'stepping stones' for benthic life[9], that has led to the development of a distinctive modern-day Southern Ocean species distribution and diversity.

South Georgia is the largest of the Atlantic-Pacific sub-Antarctic islands, lying south of the Antarctic Polar Front (APF) (Fig. 1a,b). Isolated in the Southern Ocean, South Georgia is considered as a 'sentinel' of change: an area sensitive to regional climate, where future environmental change would be first detected. Glacier change is currently dramatic on South Georgia, with significant retreat recorded in >90% of its glaciers over the past $\sim$60 years[10,11]. Observations that current and former ice-cap configurations were marine-terminating, glacier systems exhibit rapid throughflow and thus respond quickly to environmental changes[10], and the island's maritime climate reflects regional forcings, make it equally likely that past environmental change on South Georgia was highly sensitive to wider climatic influences. Specifically, complex ocean water mass circulation influences the island's climate (Fig. 1), which is susceptible to latitudinal variations in the position of the APF, and to changes in the intensity and location of the Southern Westerly Winds whose core belt (50–60 °S) supplies significant amounts of moisture to the island. It is reasonable to assume, therefore, that glacier behaviour on the island was strongly coupled to Southern Hemisphere (Antarctic) climate variability in the past, as it is today. It follows that an improved understanding of South Georgia's glaciations can shed light on ice mass response to climate variability in an under-sampled but regionally-important Southern Ocean sector, and can provide as yet unknown long-timescale context for changes in the sub-Antarctic cryosphere over recent decades.

Despite being the most well studied of the sub-Antarctic islands the number and timing of past glaciations on South Georgia remains unclear[12]. Lagging significantly behind reconstructions of palaeo-ice sheets in the mid-latitudes[13–16] and at the poles[6,17,18]; we do not even have an understanding of the maximum ice-sheet configuration at the Last Glacial Maximum (LGM; 19–26 ka), let alone the subsequent deglacial history. Previous studies have inferred two conflicting LGM models: the first inferring extensive shelf-wide glaciation and subsequent retreat to present limits[19]; the second suggesting ice-cap extents restricted to near-coastal[20] or inner-fjord[21] limits, with cross-shelf troughs and offshore sea-floor geomorphology mapped from coarsely-gridded regional bathymetry[22] relating to older, more expansive preceding glaciations. While most investigations to-date have favoured the latter restricted hypothesis, recent modelling[23] and biodiversity studies[24] have

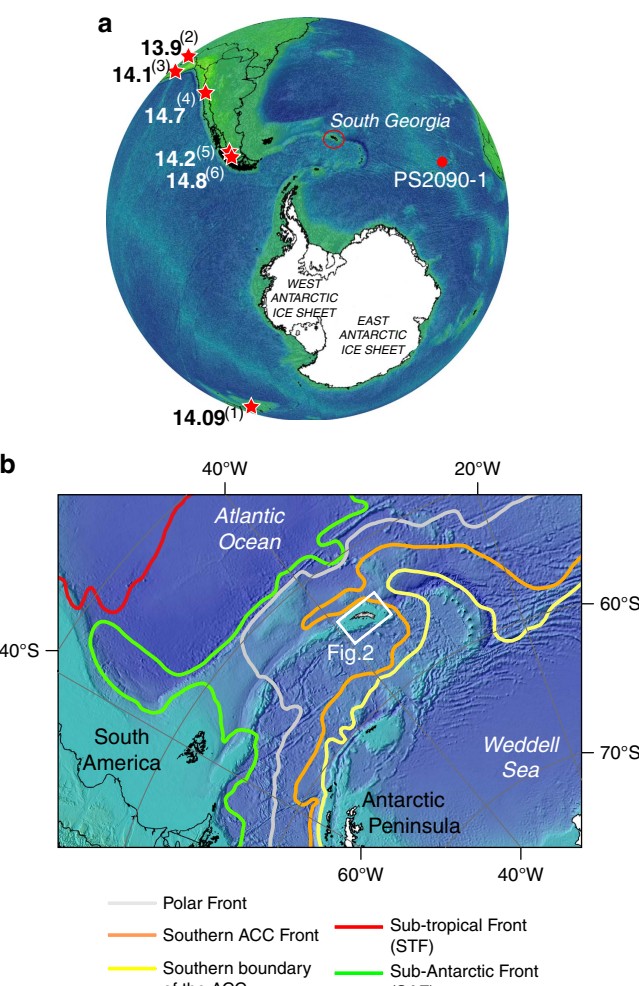

**Figure 1 | Map of South Georgia.** (**a**) Southern Hemisphere map showing sites where advances of glaciers during the Antarctic Cold Reversal have been recorded. Bold text are maximum ages (in ka B.P.) for the timing of maximum advance in each location. (1—ref. 28), (2—ref. 30), (3—ref. 31), (4—ref. 32), (5—ref. 33), (6—ref. 34). (**b**) Regional location map illustrating modern-day mean annual position of major Southern Ocean oceanographic fronts in relation to South Georgia.

both inferred a larger LGM ice-cap. Crucially, however, neither model has yet been comprehensively tested.

The debate over LGM extent is mirrored in the lack of consensus over glacier variability during the ensuing deglaciation (Termination 1). Rosqvist et al.[25] suggested that deglaciation of the island commenced before $\sim$18.6 ka B.P.[26] based on the onset of accumulation of lake sediments at Tønsberg Point. The authors' radiocarbon dates have since formed key evidence used to argue in favour of a restricted LGM hypothesis suggesting ice-free conditions on land at the LGM. Other studies have dated deglaciation from the onset of peat and lake sediments as significantly younger, for example at Dartmouth Point, where a minimum age for deglaciation is constrained to $\sim$10.8–10.2 cal ka B.P. (ref. 27). A notable period absent in most geological records from South Georgia is the phase of warming that brought about the deglaciation of Antarctica, from $\sim$18 ka to the start of the Holocene. During this deglaciation conditions were interrupted by a period of renewed cooling from about 14,540 to 12,760 kyr ago[28]. This chronozone, known as the Antarctic Cold Reversal (ACR), is documented in Antarctic ice

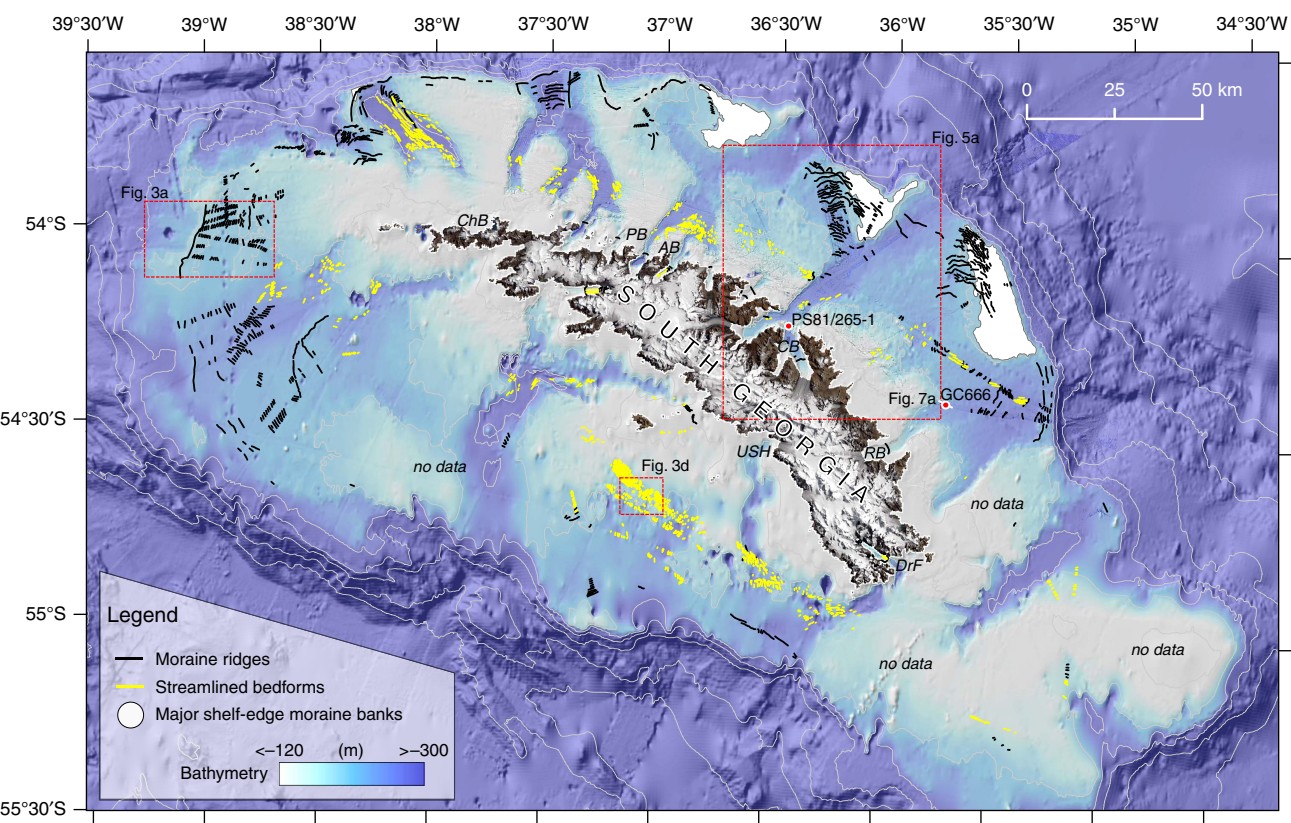

**Figure 2 | Overview of sea-floor landforms mapped in this study.** Bathymetry of the South Georgia continental block (ref. 22), with newly mapped moraine ridges (black), streamlined bedforms (yellow) and moraine banks (light grey) depicting the imprint of an entire sub-polar palaeo-ice cap. Locations of subsequent figures shown by red boxes. ChB: Church Bay, PB: Possession Bay, AB: Antarctic Bay, CB: Cumberland Bay, RB: Royal Bay, DrF: Drygalski Fjord, USH: Undine South Harbour.

cores, but understanding of the spatial extent of this cooling in the Southern Hemisphere is limited to only a handful of sites[29]. Glacier advances have been reported from the tropical Andes[30,31], Patagonia[32–34] and New Zealand[28], but the spatial footprint of glacier response to the ACR, at South Georgia and other Southern Ocean islands, is not known (Fig. 1a). A critical reason for the lack of progress in determining the glacial evolution, including the LGM extent and dynamics of the last termination, is a dearth of geophysical and stratigraphic data from the marine environment[12], where changes through these intervals are most likely to be recorded[21].

This study uses sea-floor geophysical data and new marine sediment cores to constrain the glacial history of South Georgia from a unique offshore perspective. Multibeam swath bathymetry, acoustic sub-bottom profiler data, and gravity cores were collected on two recent cruises to the island: JR257 (RRS *James Clark Ross*) in 2012, and ANT-XXIX/4 (RV *Polarstern* PS81) in 2013. We use these data to firstly map and analyse glacial features from the shelf sea-floor to reveal landforms recording former ice-cap flow and extent. Second, we use two sediment cores from the glacial troughs to constrain the landform record, and document changes in the sedimentary environment. Our findings reveal an extensive last glaciation during which an expanded ice cap covered a substantial proportion of the continental block, and a major readvance into the fjords during the ACR.

## Results

**Landform observations from multi-beam echo-sounder data.** We combined and analysed newly-acquired and existing sea-floor

bathymetric datasets to reveal the imprint of former glaciations for an entire sub-polar ice cap on South Georgia (Supplementary Fig. 1). Figure 2 shows a detailed map of glacial landforms on the continental block. In total, we have mapped >770 moraine ridges and >1,750 streamlined landforms which we interpret were formed at the margins of, and beneath, former expanded grounded ice caps, respectively.

At a broad scale, the new data set shows systematic patterns of submarine glacial features indicative of multiple former glaciations of the shelf (Fig. 2, Supplementary Fig. 2). Large outer shelf banks, 50–120 m in relief and 10–20 km wide, are interpreted, based on their morphology, as moraine banks, and occur in association with at least three of the cross-shelf troughs north of the island, flanking the troughs to their west. These banks are mantled by successions of smaller amplitude moraine ridges recording former ice limits, with the largest and most continuous found at or near to the shelf break in almost all areas for which we have data coverage (Fig. 2). The orientation of the smaller ridges is consistently sub-parallel to the axes of the main troughs, and cluster within northwardly-trending secondary troughs that clearly offshoot the primary (Supplementary Fig. 2). Lobate geometries characterize the termini of these secondary troughs, and are consistent morphologically with the switching of palaeo-ice flow within, or between, glacial periods[35]. Only a few moraine ridges occur within the primary troughs themselves (Fig. 2). Trough dimensions with excavation of several 10 s m can only be explained as long-term phenomena, while secondary troughs are graded to a similar depth, and display comparable landform assemblages from the most recent retreat, all of which implies a common history of landscape development. We suggest the

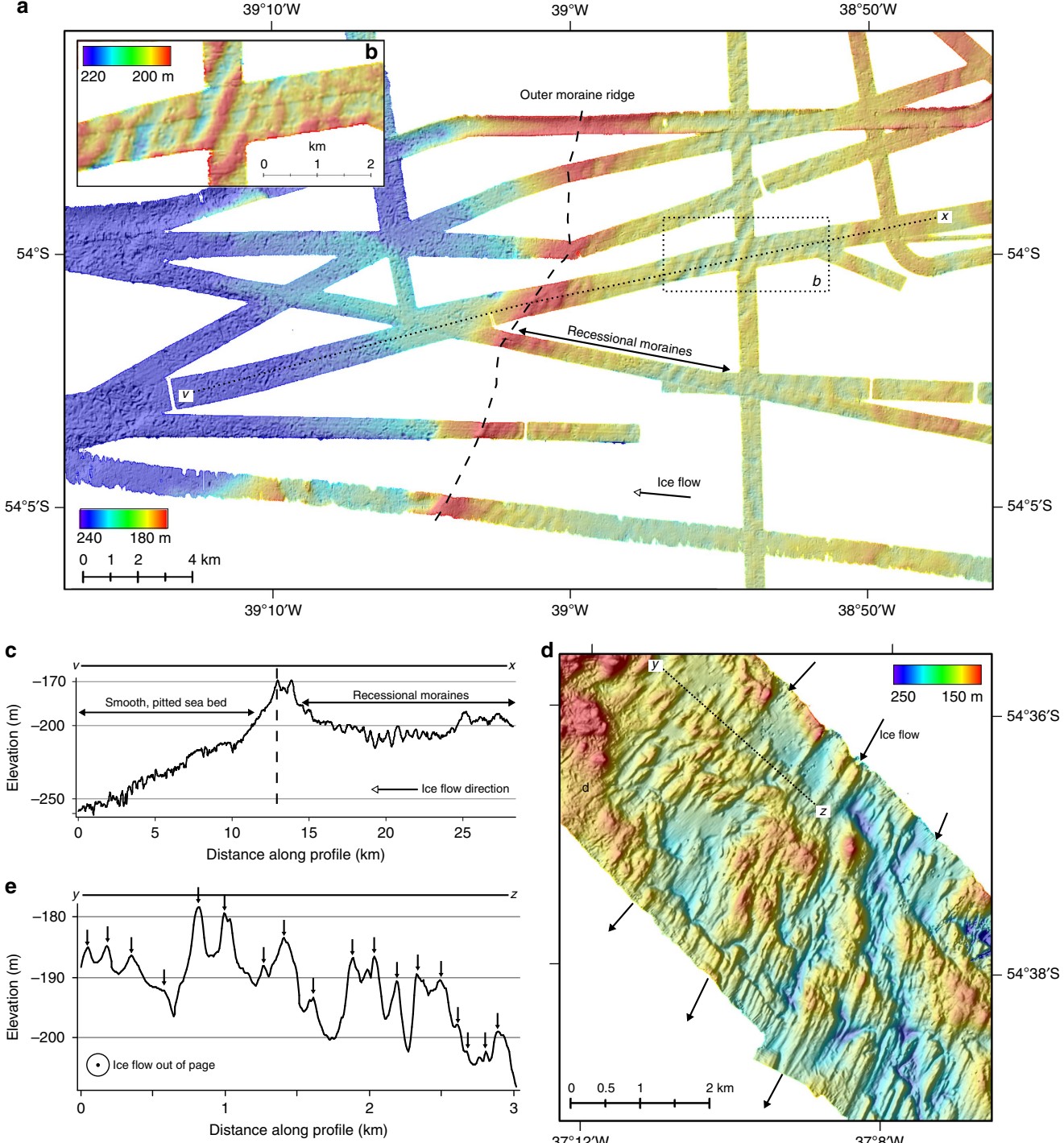

**Figure 3 | Detailed multibeam swath bathymetry maps showing glacially-formed landforms.** (**a**) Compiled swath bathymetry tracks west of the island illustrating a large outer ridge and back-stepping recessional moraines; profile *v–x* through the ridge shown in '**c**'; (**b**) a sub-set of the recessional moraines in detail. (**c**) profile *v–x* through the outer ridge showing the separation of deeper, pitted sea-floor and moraine-mantled topography inshore. (**d**) streamlined subglacial bedforms south of the island, resembling drumlins formed by warm-based, fast glacial flow. (**e**) topographic profile *y–z* across the bedforms illustrating their range of amplitudes and typical wavelengths. Profile located in '**d**'. Data in '**a**' gridded at 5 m grid cell size, and at 8 m grid-cell size in '**d**'.

scale and complexity of this shelf morphology trough-to-trough can only be explained as a product of long-term development as a consequence of repeated glaciations that frequently reached outer-shelf limits.

West of the island, swath bathymetry datasets image in more detail the distinct outer moraine ridge (~30 m high, 4 km wide)

and corresponding suite of sub-parallel smaller sharper-crested ridges inshore (2–15 m high, 200–600 m wide) that we interpret as recessional ice-marginal moraines[36–38] (Fig. 3a–c). Beyond the outer ridge, water depths increase gradually seaward, where the sea floor is pitted (possibly fluid or gas-escape pockmarks[39]) but otherwise featureless (Fig. 3c). It is clear that there is no sea-bed

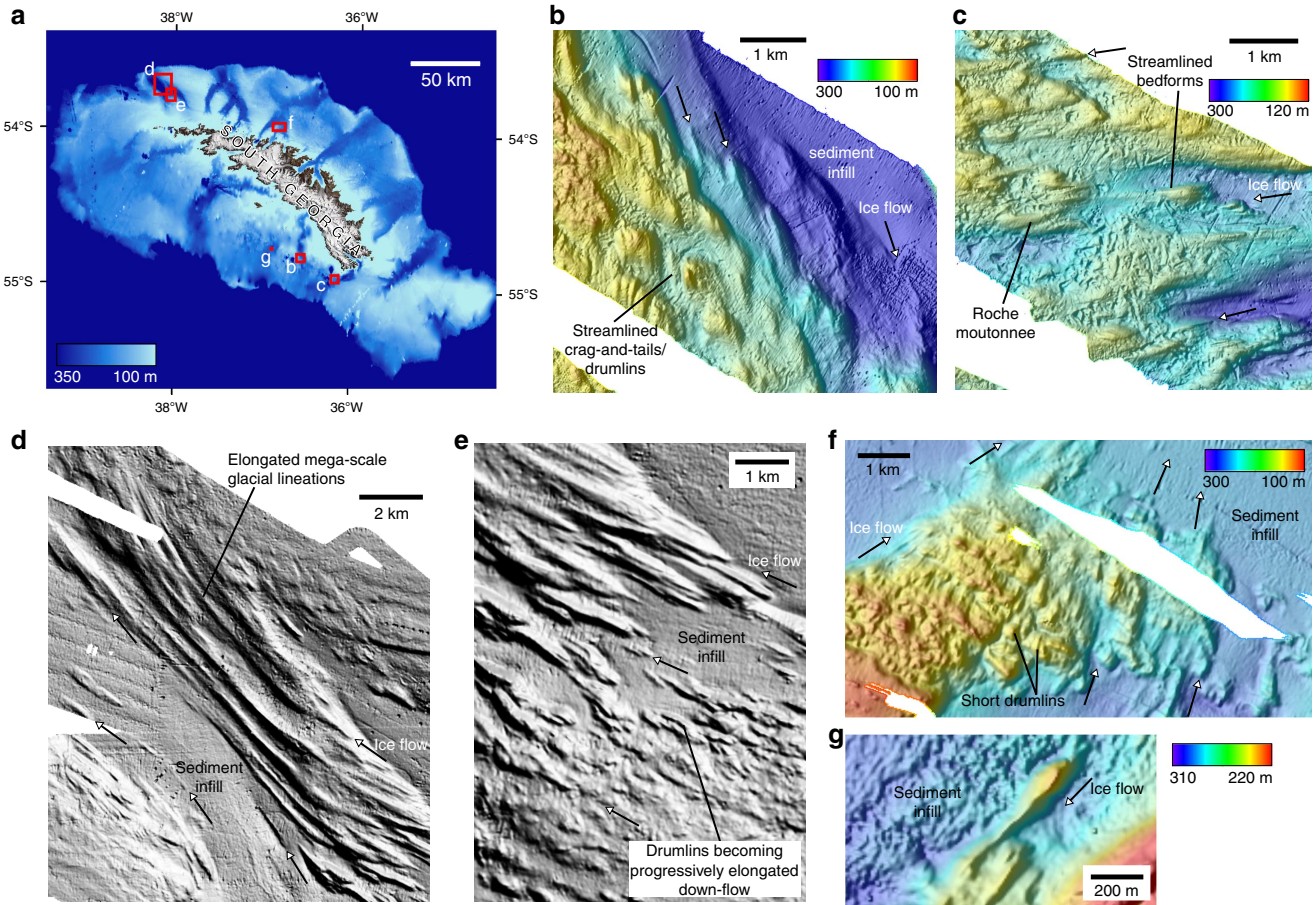

**Figure 4 | Multibeam swath bathymetry maps of streamlined subglacial bedforms on the South Georgia shelf.** (**a**) Location map indicating the locations of panels **b**–**g**. (**b**) Streamlined crag-and-tails and/or drumlins 0.5–1 km in length, south of Undine South Harbour; (**c**) streamlined bedforms and roche moutonnée overprinted by extensive scouring in the trough feeding out of Drygalski Fjord; (**d**) elongated mega-scale glacial lineations and (**e**) drumlinoid bedforms that increase in elongation down-flow in the trough northwest of Church Bay; (**f**) convergence of flow patterns including a swarm of small drumlins at the confluence of tributaries emanating from Possession Bay and Antarctic Bay; (**g**) individual drumlin imaged on flat sea-bed region south of the central part of the island. All multibeam data have an 8 m grid-cell size, except for '**f**' which is gridded at 20-m cell size. Arrows depict direction of former ice flow.

imprint of ice grounding beyond this limit, and thus we have confidence in the outermost ridge marking a maximum limit of former glaciations.

Subglacial streamlined bedforms occur within and at the flanks of many of the cross-shelf troughs (Fig. 2), and are also mapped across zones of flatter sea-floor topography south of the island (Figs 3d and 4). Bedforms are imaged in areas of little overlying sediment cover, and range in amplitude from ~1 to 15 m, have wavelengths on the order of 100–250 m, with lengths of 150–1,000 m providing elongations of ~2–10:1 in one sub-set of the data we studied (Fig. 3d,e). On the floor of many of the troughs bedforms are likely present, but are covered by sedimentary infills that have presumably accumulated through Holocene and pre-Holocene times (for example, Fig. 4a,f). Viewed across a number of examples (Fig. 4), bedforms exhibit the characteristics of drumlins (Fig. 4b,g), roche moutonées (Fig. 4c) and occasionally mega-scale glacial lineations that have commonly been associated with zones of past ice-stream flow on the Antarctic shelf (Fig. 4d)[40–50]. Convergent patterns of streamlined bedforms that become progressively more elongated down-flow (for example, Fig. 4d–f) indicate that former configurations of the ice cap contained faster-flowing sectors[51], analogous to, but forming smaller equivalents of, ice streams in Antarctica today. Some zones of the streamlined topography are

not constricted to troughs and suggest that fast flow may have been laterally-extensive near the margins of the ice cap. Other sets of bedforms, for example those in the trough NW of Church Bay (Figs 2 and 4d), are highly-elongated lineations (>10:1) and extend across the entire shelf. On the basis of observations of modern ice-stream beds[52], these can be linked with some certainty to faster-flowing ice tributaries. In Arctic ice caps, today, isolated outlet glaciers are shown to flow at speeds 7–10 times faster than that of the surrounding ice[53]. By analogy to modern ice-cap surface morphology, and to reconstructed[54] and modelled surface profiles of the Antarctic ice sheets at glacial maxima[55], we thus interpret outer portions of the ice cap to have had fast-flowing concave 'low-slung' profiles, as opposed to convex surfaces normally associated with the margins of domed slower-flowing ice caps[53]. In turn, we suggest that many outlets were only lightly grounded at thicknesses close to the threshold of flotation, with corresponding low-elevation lines of equilibrium.

For one system north of South Georgia, we have completely mapped the sequence of moraine ridges extending from the tidewater front of Nordenskjöld Glacier to the shelf break (Fig. 5a). New multibeam swath bathymetry data were collected in 2013 in a northward branching outlet of the Cumberland Trough (Fig. 5b), which extends for ~70 km from coast to

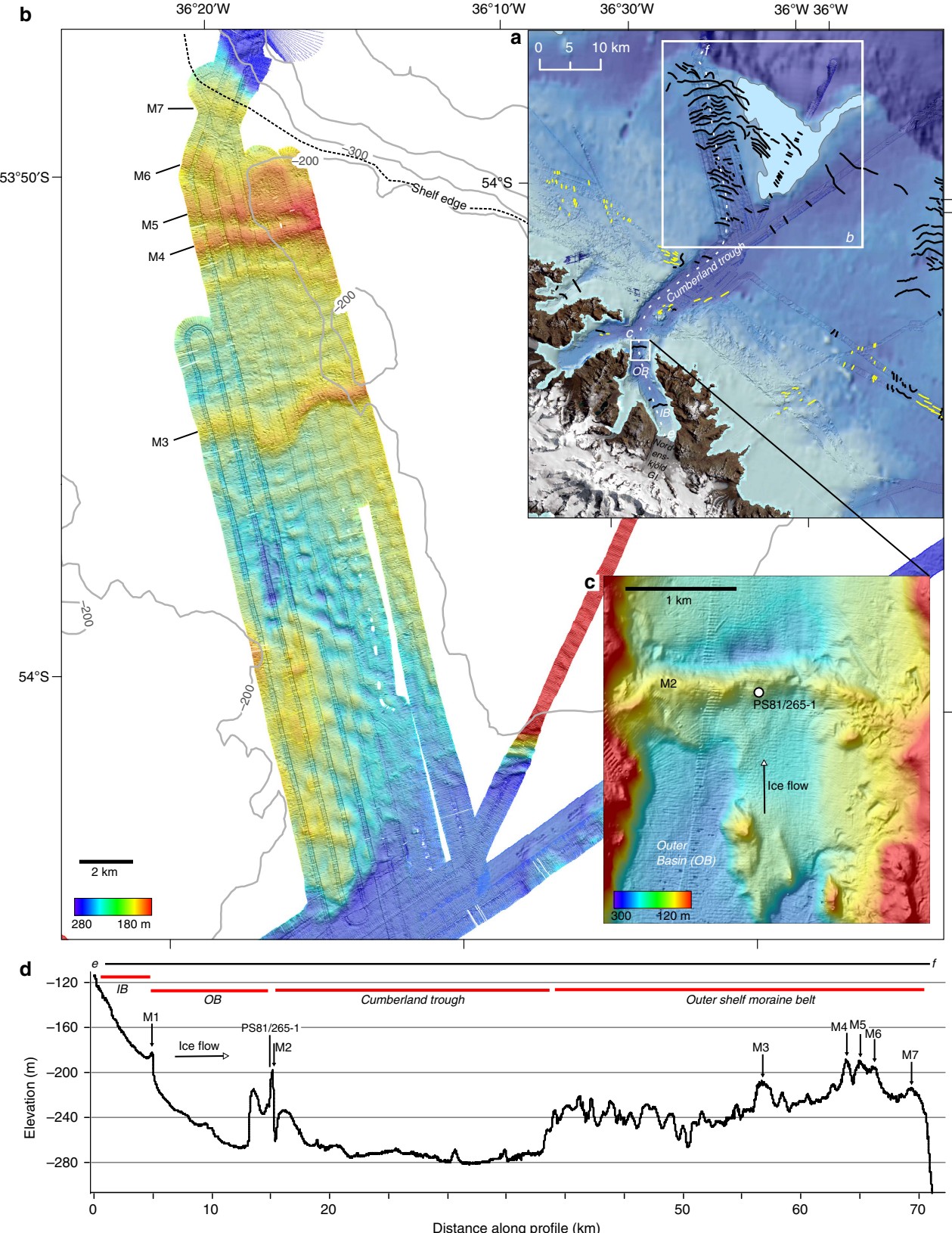

**Figure 5 | Atlas Hydrosweep DS-3 multibeam swath bathymetry collected in the secondary trough north of Cumberland Trough.** (**a**) Location of the dataset with an overview of mapped bedforms in the system. (**b**) Hillshaded swath bathymetry showing an abundance of sea-bed moraines. Major moraine ridges (M) numbered (M3–M5 interpreted as the LGM extent). (**c**) Kongsberg EM710 multibeam swath bathymetry data from the mouth of Cumberland Bay, showing the cross-trough moraine M2 and the location of a core (PS81/265-1) that targeted the moraine sediments. M2 is interpreted to correspond to the Antarctic Cold Reversal advance. (**d**) Topographic profile from the Nordenskjöld Glacier terminus to the outer shelf showing the details of the geomorphology along the former flow line. IB, inner basin; OB, outer basin.

margin in its entirety (Fig. 5d). In this location we observe a suite of pristine (unweathered) and sub-parallel ridges forming a large 'outer shelf moraine belt' (Fig. 5b,d—prominent moraine features marked and numbered from the coast seaward). The largest moraine occurs once again near to the shelf break, but is associated with more than one ridge crest (M4 and M5) and clearly dissects one or more arcuate, older moraines (M6). A smaller ~10-m high ridge also forms a 'lip' to the shelf edge (M7; Fig. 5d).

The presence of a number of significant cross-cutting ridges on the outer shelf provides support for more than one ice advance to the shelf edge. The 'outer shelf moraine belt' is mainly made up of smaller, lower amplitude ridges landward of M4 although a pronounced 30-m high ridge occurs on the mid-to-outer shelf (M3). Significantly, the ridge set as a whole arcs progressively to the south back into Cumberland Trough, indicating that the majority of ridges track the gradual evolution of a single receding ice margin through time.

Two moraines (M1 and M2) form the most prominent features of the Cumberland Bay fjord, marking the boundaries of a shallow inner (IB) and deep outer sedimentary basin (OB), respectively (Fig. 5a,d). Both basins are sediment filled, as typical for mid-to-high latitude fjords where postglacial and modern accumulation rates are exceptionally high. This is interpreted to be a result of their proximity to tidewater glaciers; in ice-proximal settings in Alaska, sedimentation rates reach metres per year, while in comparable fjords with more modest detrital input in East Greenland, Holocene sediments accumulated at rates of 110–340 cm kyr$^{-1}$ (ref. 56). M2 resides at the mouth of Cumberland Bay, >16 km from the present-day ice margin (Fig. 5c) adjacent to the farthest extents of the island's northward-jutting peninsulas. It is 30–60 m high and forms a distinct glacial limit across the breadth of the fjord floor. A comparably positioned and sized moraine is known to exist at the mouth of the majority of major fjord outlets around South Georgia[21] forming part of a consistent fjord-basin-moraine sequence that repeats from location to location (Fig. 2; Supplementary Fig. 3).

**Sediment cores and chronology.** Core PS81/265-1 was collected through moraine M2 immediately south of its crest and recovered a sedimentary sequence comprising three lithological units (Figs 5c,d and 6a). Facies analysis (Fig. 6a–d) and physical properties data (Fig. 6e), together with x-radiographs (Fig. 6i) confirm that the gravity core sampled a massive sandy diamicton at its base (2.55–3.74 m; Unit I). The presence of angular to sub-angular clasts (diameter up to 9 cm) and homogeneity of the surrounding muddy matrix indicates that the unit constitutes a sample of the body of the moraine ridge itself (Fig. 6j). It is interpreted to have formed by sediment delivery at and beneath a tidewater glacier margin. A similar sediment recovered from moraine units in front of Kongsvegen, Svalbard, was interpreted as a combination of supraglacial and englacially-thrusted basal debris[57].

The overlying unit between 2.05 and 2.55 m (Unit II) is a sandy gravelly mud with muddy inter-beds indicating ice-proximal glacimarine sedimentation, typical for grounding-line fan or subaqueous glacifluvial systems[58], that represents a transition from glacial marginal to distal glacimarine conditions. Stratification in the lower part of the unit is interpreted as a result of melt out of debris near to and away from the ice margin or the runout of debris flows sourced subglacially (Fig. 6c). Gradual reduction in stratification, clast content and shear strength upward within the unit indicates an increasing dominance of deposition of suspended detritus as the ice-margin retreated from the core site. This gradual

pull-away of the glacier is supported by steadily declining Illite/Chlorite ratios through the unit, which show a transition from high chlorite content in the underlying diamicton to low chlorite concentration in the overlying open marine facies (Fig. 6g). This change in clay mineral assemblage is characteristic of similar transitional sequences recording ice-sheet deglaciation on the West Antarctic shelf[59,60]. The upper ~2 m of the core (Unit III) is a bioturbated diatom-bearing mud interbedded with discrete 1–20 cm thick gravel layers. Unit III records open marine sedimentation in an ice-distal environment with phases of iceberg rafting, debris flows, or possibly minor glacier readvances recorded by the coarse-grained sub-units.

To constrain the age of the M2 moraine at the fjord mouth we dated shell fragments recovered from the matrix of the moraine unit (Unit I at 3.055 m) (Fig. 6a). We suggest that the presence of shell material within the diamicton can only be explained by the reworking of existing fjord sediments into the ice-basal material, that were deposited in the fjord prior to ice advance, or directly at the margin of a tidewater glacier. There is no evidence for burrowing in x-ray images, so the shell is interpreted to be related to the primary deposit that has subsequently been reworked. Thus, we interpret the moraine as the product of a frontal advance rather than as a feature formed during a pattern of overall ice retreat. Following similar studies that used shell fragments as indicators for the age of glacial sediments[61], the age of the shell material provides a reliable age for the formation of the moraine ridge, suggesting it was deposited sometime after 14.8–15.4 cal ka B.P. (Table 1). A second age from the very top of Unit I (2.56–2.60 m; 9,891 cal yr B.P.) is excluded from the core age-model as an outlier; the only one of five radiocarbon ages that otherwise lie in stratigraphic order. The anomalously young age is interpreted as the result of dissolution in the sample[62], with chamber damage and external corrosion evident on some of the largest foraminiferal specimens.

The overlying transitional unit (II) is reliably dated in two places using foraminifera (Fig. 6a,b). An age from the base of the unit shows the transition from subglacial to ice-proximal sedimentation to have occurred immediately before 13,340 cal yr B.P (Fig. 6). A further date from the unit indicates nearby glacier sources until at least 10,637 cal yr B.P. with a reducing glacial influence thereafter. The boundary between the base of the open marine Unit III and very top of the transitional unit II in PS81/265-1 was dated to 2,228 cal yr B.P. revealing that the onset of open marine conditions is surprisingly late. To explain this, we suggest a hiatus between retreat from the moraine and the onset of marine conditions at the core site. This break in sedimentation likely resulted from sediment focussing in the basin behind the moraine, and subsequent infill. In other basins offshore South Georgia, sediments clearly accumulate at faster rates in depressions compared with moraine crests (Fig. 7a, arrowed), which implies that breaks in the continuity of the sedimentary record can be expected in areas where the underlying till or bedrock shallow. Thus, it is suggested that the onlap of marine sediments at PS81/265-1 initiated in the late Holocene, well after deglaciation inshore.

Gravity core GC666 was collected in Royal Bay Trough, an adjacent glacier system southeast of Cumberland Bay (Fig. 2). The Royal Bay Trough is up to ~260 m deep and extends from the Ross-Hindle glaciers northeastwards. Sub-bottom profiler data through the core location show a ~20 m thick stratified sedimentary sequence within the trough that overlies transparent acoustic units interpreted as either glacial diamictons or sub-cropping lithified sediments/bedrock (Fig. 7a). GC666 recovered ~8 m of diatomaceous muds to the west of the trough axis (Fig. 7b–h). The core is comprised of two lithological units.

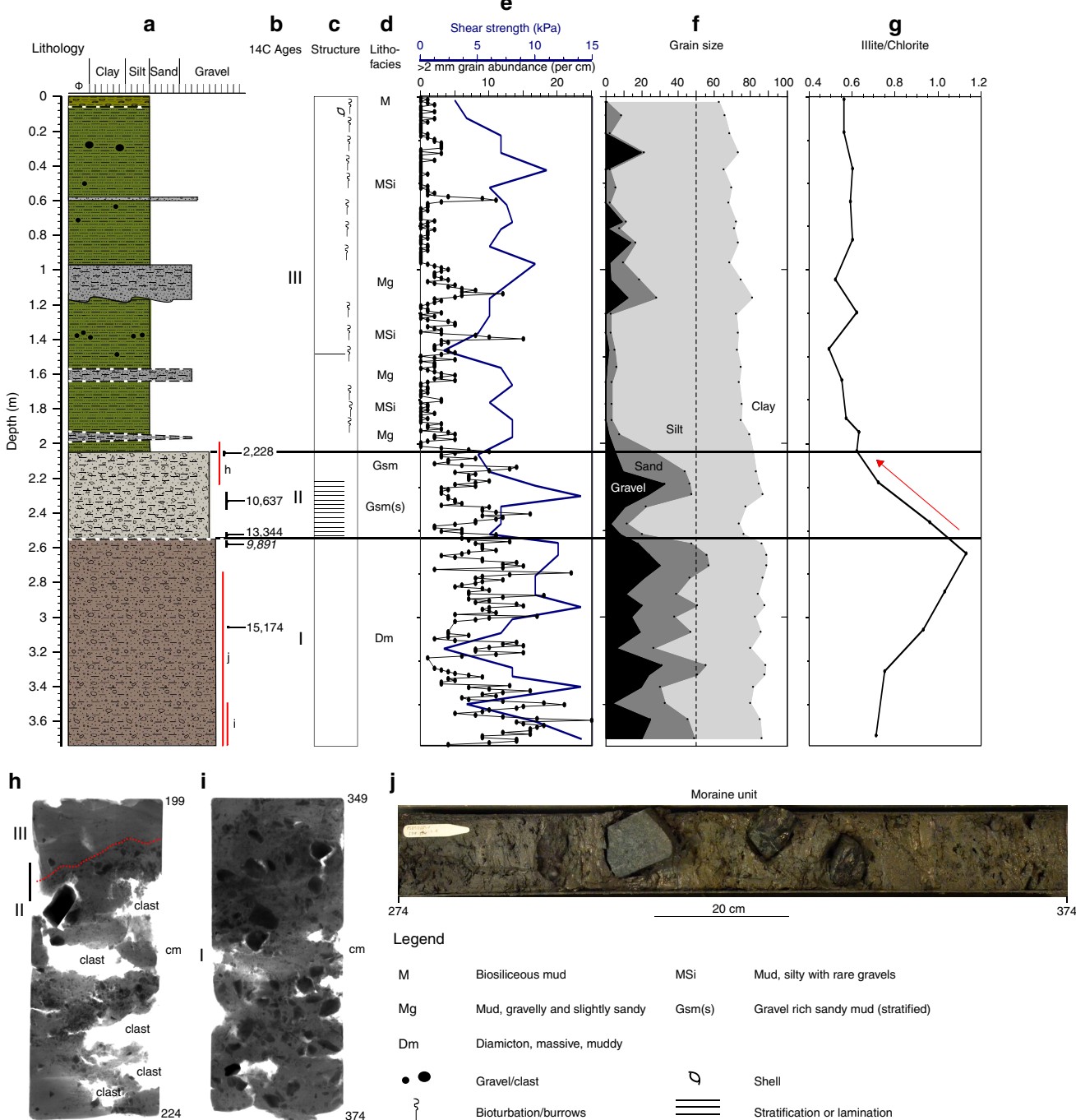

**Figure 6 | Sediment logs and properties for core PS81/265-1 through moraine M2.** (**a**) Lithological log showing key units and dominant grain size. Colours used to distinguish units and are representative, (**b**) AMS 14C ages, (**c**) sedimentary structures, (**d**) lithofacies, (**e**) shear strength data (blue), grain abundance >2 mm per 10 cm³ (black dots), (**f**) grain size percentages, and (**g**) the Illite/Chlorite ratio in the clay (<2 μm fraction). (**h,i**) X-radiographs showing massive structure of the gravel-rich units. Black bar in '**h**' shows location of sample for dating, red line shows boundary between units (**j**) core photograph of a section of the lowermost 'moraine unit' (lithological Unit I) illustrating large angular clasts in homogenous matrix. Legend relates to panels **a–d** only.

Lower Unit I consists of laminated diatomaceous oozes alternating with fine-grained, more terrigenous diatom-rich muds, interpreted to record inter-annual or even seasonal deposition in an open marine setting. Individual laminae can be detected in the variability recorded by core wet-bulk density data (Fig. 7e). Radiocarbon ages at the base of Unit I indicate the onset of marine sedimentation at the site before ∼14.6–15.1 cal ka BP. Two ages of ∼14.0 and 14.1 cal ka B.P. near the top of Unit I

suggest rapid sedimentation rates through the lower part of the core (Fig. 7i). By contrast, lithological Unit II (above ∼4.6 m) is a homogeneous diatomaceous mud, with a higher terrigenous sediment fraction. Radiocarbon dates document its deposition at sedimentation rates lower than those for Unit I.

Although the sedimentary record in GC666 suggests open marine conditions throughout, a distinct facies change can be seen in acoustic profiles, which corresponds to the change in

**Table 1 | Radiocarbon age determinations for cores PS81/265-1 and GC666, from the South Georgia shelf.**

| Sample details | | | Conventional ages | | | Marine13 calibration | | | |
|---|---|---|---|---|---|---|---|---|---|
| Core | Depth (cm) | Carbonate Source | Age (yr) | ± Age Error (yr) | Reservoir Effect (yr) | Min (cal yr BP) | Max (cal yr BP) | Median (cal yr BP) | Lab Code |
| GC666 | 0 | benthic forams | 1,186 | 65 | 1,108 ± 61 | 0 | 266 | **124** | ETH-51517.1 |
| GC666 | 0 | benthic forams | 1,030 | 56 | 1,108 ± 61 | 0 | 259 | **122** | ETH-51517.2 |
| GC666 | 388 | gastropod | 11,926 | 80 | 1,108 ± 61 | 12,557 | 12,912 | **12,712** | ETH-51518.1 |
| GC666 | 388 | gastropod | 11,726 | 109 | 1,108 ± 61 | 12,114 | 12,756 | **12,527** | ETH-51518.2 |
| GC666 | 490 | benthic forams | 13,370 | 89 | 1,108 ± 61 | 13,885 | 14,758 | **14,231** | ETH-51519.1 |
| GC666 | 490 | benthic forams | 13,105 | 136 | 1,108 ± 61 | 13,477 | 14,226 | **13,874** | ETH-51519.2 |
| GC666 | 510 | shell fragments | 13,301 | 135 | 1,108 ± 61 | 13,734 | 14,799 | **14,145** | ETH-51520 |
| GC666 | 510 | benthic forams | 12,005 | 275 | 1,108 ± 61 | 12,275 | 13,455 | **12,865** | ETH-51521 |
| GC666 | 815 | benthic forams | 13,572 | 211 | 1,108 ± 61 | 13,918 | 15,321 | **14,628** | ETH-51522 |
| GC666 | 815 | planktic forams | 13,896 | 492 | 1,108 ± 61 | 13,702 | 16,589 | **15,143** | ETH-51523 |
| PS81/265-1 | 204–207 | benthic forams | 3,270 | 30 | 1,108 ± 61 | 2,040 | 2,392 | **2,228** | BETA-402960 |
| PS81/265-1 | 228–238 | benthic forams | 10,440 | 40 | 1,108 ± 61 | 10,437 | 10,867 | **10,637** | BETA-449408 |
| PS81/265-1 | 251–254 | benthic forams | 12,590 | 40 | 1,108 ± 61 | 13,195 | 13,491 | **13,344** | BETA-444223 |
| PS81/265-1 | 256–260 | benthic forams | 9,840 | 40 | 1,108 ± 61 | 9,645 | 10,140 | **9,891** | BETA-444224 |
| PS81/265-1 | 305.5 | gastropod | 13,850 | 40 | 1,108 ± 61 | 14,857 | 15,432 | **15,174** | BETA-402961 |

Bold text highlights the median age used in the text.

lithological units down-core (Fig. 7a,i). An initially biologically-productive glacially-influenced marine environment characterized by high sedimentation rates (Unit I) was replaced by a less productive hemipelagic shelf environment (Unit II). Higher terrigenous sediment content in the upper unit may be due to a dilution effect, rather than an indicator of supply. A regional break or unconformity forms the boundary between the two units, at ~4.6 m below sea floor, in acoustic profiler data and marks the main change in depositional environment as well as an erosional event that removed sediments from the trough. We interpret this change to correspond principally to the degree of glacial influence (that is, to a decrease in glacier proximity and thus reduction in productivity across the boundary, together with a short-lived phase of marine erosion). Above the unconformity, acoustic facies remain weakly stratified. Based on radiocarbon dates below the corresponding boundary in GC666, the prominent change in deposition likely occurred shortly after ~14 ka B.P. Rates of deposition were moderate from ~14.1 ka B.P. but had slowed considerably by, and after, 12.6 ka B.P.

## Discussion

Today, a significant sea-floor landform and sedimentary record of past ice-cap change is preserved on the South Georgia continental block. Within this record, there is evidence for multiple extensive glaciations of the island. This evidence takes the form of: (i) submarine morainal banks, overprinted by stratigraphically younger moraine ridges, (ii) diverging trough systems with repeated complex geomorphic patterning between adjacent systems, suggesting a common history of glacier modification (Supplementary Fig. 2), (iii) shelf-edge ridges truncated by further moraines created by subsequent glacier advances (for example, Fig. 5; M6 and M7). Nevertheless, if the sea-bed geomorphology inshore of the shelf edge recorded multiple glaciations (cf. the moraine sequences onshore at the margins of the Patagonian Ice Sheet[63–66]) we would expect to observe a network of overprinting and cross-cutting ridges comprising of a mixture of degraded and fresher moraines. Instead, in almost all locations we have imaged only one distinct outer moraine ridge and a suite of smaller recessional moraines (Figs 2, 3a and 5b). The continuity of forms, the lack of degradation on the ridge surfaces, and their modern-day sea-floor exposure (they are neither buried nor heavily eroded) suggests strongly that the latter record,

predominantly, a single phase of recent ice cap deglaciation[38]. Two previous models were proposed for LGM extent on South Georgia and the majority have favoured a restricted interpretation[20,21,25]. Contrary to most studies, our observations suggest that the mapped limits of moraines around the island relate to a 'recent' advance-retreat cycle, which we propose corresponds to expansion at the Last Glacial Maximum and the ensuing deglaciation.

If the 'outer moraine belt' relates to the LGM ice advance and deglaciation, then one would expect the quantity of subsequent trough sediment infill to be consistent with expected rates of accumulation over that timeframe. To test the above hypothesis of extensive shelf glaciation, we used sedimentation rates calculated from radiocarbon determinations on GC666 to estimate the age of the entire trough infill (Fig. 7c; Table 2). Two slightly different core basal ages suggest rates of deposition ranging between 306 and 632 cm kyr$^{-1}$ for the period corresponding to the latter part of the last termination (Fig. 7i). GC666 only recovered the upper 8 m of c. 20 m of trough deposits, but acoustic facies suggest that Unit I extends from the core base to the trough floor and that the depositional regime has remained unchanged (Fig. 7a). If Unit I relates to marine sedimentation during deglaciation, as seems likely from our dating, it would be plausible that sedimentation rates remained high for the duration of the period. Thus, we are confident in extrapolating these high-sedimentation rates downwards through the section. Even with the most conservative rates of deposition, we estimate that trough sediments began accumulating ~17,900 yr ago (Fig. 7c). On the inner parts of Antarctic ice-stream troughs, deglacial-to-Holocene sediments overlie subglacial diamicton or bedrock directly, with ice having stripped away pre-existing fill, advecting this sediment to the margin[49,67,68]. The apparent absence of older deposits within the mapped troughs suggests that the South Georgia shelf records a similar shallow stratigraphic architecture to many Antarctic glacial systems. Thus, combined dating and geophysical analysis supports the interpretation of ice advance beyond the fjord mouths at the peak of the last glaciation, and onset of trough sediment fill at the beginning of the last deglaciation.

To provide an independent chronological test of LGM extent beyond the coastal limits, we dated sediments recovered from moraine M2 at the mouth of Cumberland Bay. According to our previously posed model (limited extent model), this moraine should date to pre-LGM times, possibly having

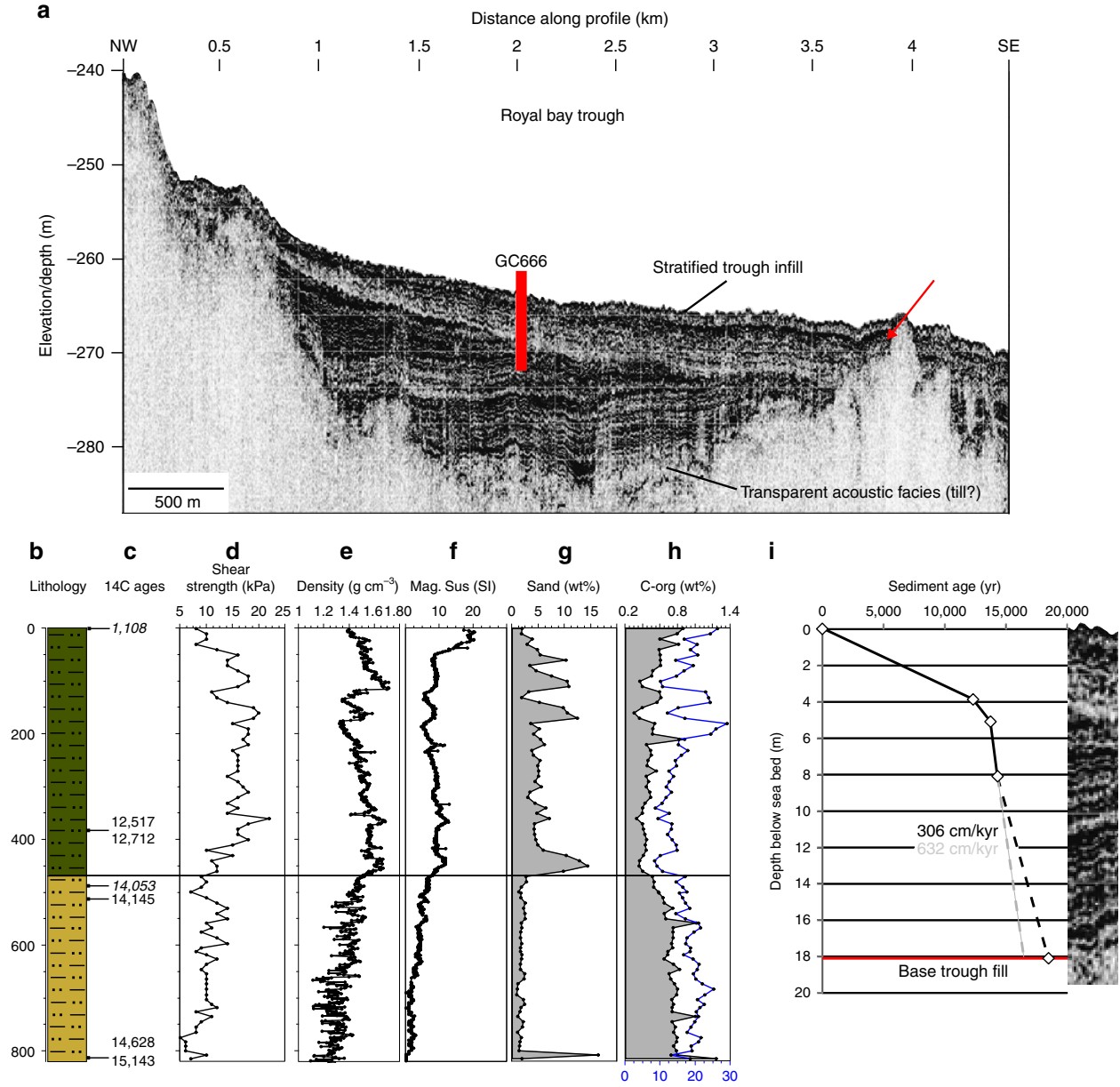

**Figure 7 | Characteristics of trough fill sediments in Royal Bay at core site GC666.** (**a**) Atlas PARASOUND acoustic sub-bottom profile across the GC666 coring site (red bar). The ∼20-m of stratified trough infill onlaps a transparent underlying acoustic facies. At shallow locations the lower units are not represented (for example, red arrow). Core log and properties for gravity core GC666, showing (**b**) simplified lithology, (**c**) AMS [14]C ages (calibrated yr), (**d**) shear strength, (**e**) density, (**f**) magnetic susceptibility, (**g**) percentage sand content, (**h**) total organic carbon and biogenic opal content. (**i**) Age-depth model for the core showing the change in sedimentation rates at the boundary between lithological units I and II and down-section extrapolations to predict the timing of onset of trough infill, based on correlation to the PARASOUND profiles. Velocity used for correlation 1,500 m s$^{-1}$. Note that the reported age at 490 cm is an average of two radiocarbon measurements, while the age at 510 cm uses the older of two measurements at that horizon. Core top is assumed modern (conventional 14C age shown).

formed during Marine Isotope Stage 6 (130–191 ka)[21]. Thus any [14]C dates from sediments within or directly on top of the moraine unit should give radiocarbon 'dead' ages. On the contrary, our dating clearly shows that the M2 relates to an advance after ∼15.2 cal ka B.P. An age younger than ∼15.2 ka B.P. for M2 rejects the hypothesis that the last glaciation reached only inner fjord limits. It is very unlikely that a post-LGM readvance overrode a less extensive LGM limit. More significantly, there are no major moraine limits on the continental block outside of the fjord mouths until the outer shelf moraine belt, some 35 km from the modern glacier fronts

(Fig. 5b). This implies that the LGM advance was significant and extensive. The lack of discernible erosion or geometric differences in the M3 to M5 ridges suggests that this complex marks the LGM terminus (Fig. 5b,d). If these interpretations are correct, then our findings seemingly resolve a long-standing debate concerning the extent of the LGM on South Georgia, identifying an extensive last glaciation for the first time, and confirming recent results from ice-load modelling[23] and biodiversity surveys[24] that inferred a larger LGM ice-cap on the island.

The maximum age for moraine M2 of 15.2 ka B.P. immediately pre-dates the onset of the ACR, and subsequently constrains

**Table 2 | Calculated sedimentation rates for dated intervals in core GC666, from Royal Bay Trough.**

| Depth range (cm) | Accumulation (cm) | Age range (yr) | Time period (kyr) | Sedimentation rate (cm kyr$^{-1}$) |
|---|---|---|---|---|
| 0–388 | 388 | 0–12,620 | 12.62 | 30.7 |
| 0–490 | 490 | 0–14,053 | 14.053 | 34.9 |
| 0–815 | 815 | 0–14,628 | 14.628 | 55.7 |
| 0–815 | 815 | 0–15,143 | 15.143 | 53.8 |
| 388–490 | 102 | 12620–14,053 | 1.433 | 71.2 |
| 388–510 | 122 | 12,620–14,145 | 1.525 | 80.0 |
| 510–815 | 305 | 14,145–14,628 | 0.483 | 631.5 |
| 510–815 | 305 | 14,145–15,143 | 0.998 | 305.6 |

For several core depths, two ages have been recovered and thus a range of sedimentation rates are calculated for some of the depth intervals.

a significant advance of South Georgia's glaciers to this period of cooling. A study of lake sediments adjacent to Cumberland Bay fjord revealed that climate cooled during an interval between 14.8 and 14.2 ka B.P., coincident with the ACR (ref. 25). Notably, a wetter regime was also inferred from the higher grey-scale density values and silica isotope depletion during this interval when more minerogenic deposition was input to the lake[25]. Wetter, cooler climates would provide conditions favourable for glacier advance, whilst the timing of the period is consistent with the phase of ice advance suggested by our dating. Importantly, other fjords around South Georgia contain fjord-mouth moraine ridges which, based on the common gross fjord morphology and geometry[21], are likely to be equivalents to the M2 moraine in Cumberland Bay (Fig. 2). Thus, following similar moraine classification approaches terrestrially[20] we suggest that the ACR advance was not simply restricted to just one outlet system but rather a response across the entire ice cap (Supplementary Fig. 4). The nature of the readvance during this interval can be constrained by lake sediments from Tønsberg Peninsula which indicate that any expanded ice cap had already retreated substantially from mainland promontories by ~18.6 ka B.P., likely inshore of the outer-fjord positions. We infer that the ACR readvances can, therefore, only have been restricted to discrete fjords in many places, and did not override the intervening peninsulas. We cannot be certain about the true extent of the retreat up-fjord prior to the ACR, but our interpretation that the ACR saw renewed resurgence rather than a stillstand during long-term retreat is upheld by the presence of truncated moraines at shallow depths on the fjord flanks. Previously interpreted as remnant landforms of pre-LGM glaciations[21], these can now be explained as ice retreat features formed during the early phase of post-LGM deglaciation to coastal or inner fjord limits that were subsequently dissected by fjordal re-advance during the ACR (See Fig. 4a in ref. 21). In Cumberland Bay, this implies a readvance of at least 8 km down-fjord based on the most southerly position of a set of eroded promontories (Supplementary Fig. 5).

We were also able to directly date the timing of retreat following the ACR advance from PS81/265-1 which suggests ice had ungrounded from M2 by 13.3 ka B.P. In addition, core GC666 can provide constraints on changes in the nearby ocean during and after the proposed ACR advance (Fig. 7). The core shows a switch in sedimentation shortly after ~14.0 ka B.P that is mirrored in an acoustic mid-sequence unconformity. The apparent unconformity is difficult to explain. Glacial erosional processes are ruled out given the lack of sedimentary evidence for ice grounding and absence of chronological evidence for a hiatus in the core (Fig. 7c). We suggest instead bottom current erosion associated with a latitudinal shift in oceanographic fronts might be responsible, concomittent with climate amelioration and glacier retreat from ACR advance limits. At least in the vicinity

of GC666, the event appears not to have removed a significant amount of material, meaning we can interpret downcore changes across this boundary with confidence. One plausible explanation for the high sedimentation rates observed for Unit I is that these reflect enhanced production during the transition out of the last glacial stage. A local cause of the high productivity is suggested, which we infer resulted from glacier proximity to the core site. In Antarctic troughs, analogous varved couplets consisting of diatomaceous oozes and muds were deposited during intervals of high primary productivity generated by seasonal diatom blooms within ice-proximal embayments enriched with iron-laden meltwater[54,69]. In this setting, sedimentation rates were strongly linked to the pattern of ice retreat. Accordingly, if we use the timing of the major shift in sedimentation as an indicator for glacier retreat, then 14.0 ka B.P provides a likely minimum age on ice recession from coastal waters. Thus, combining observations and dating from both cores, we suggest an ACR advance and culmination between 15.2 and 14.0 ka B.P, with grounding-line retreat from the outer-fjord moraine by 13.3 ka B.P; hence, bracketing the ACR advance on South Georgia to a ~1,900 year interval of time (Fig. 8).

If the moraine M2 corresponds to the ACR advance then the inner fjord moraine M1 must have formed during a younger ice advance, analogous to the category 'a' moraines mapped onshore in Moraine Fjord by Bentley et al.[12,20,21]. The A3 moraines are the outermost of a set of three suites of moraines forming a wide, low amplitude ridge deposited during the lateglacial. The lack of older moraines beyond on the valley sides was previously considered as evidence that these ridges relate to the LGM limit. Dated to 12.2 ± 1.5 ka (ref. 20), the A3/M1 moraine may instead represent a secondary advance or stillstand of Nordenskjöld Glacier as climate reorganized at the start of the Younger Dryas. Similar two-stage advances have been noted for New Zealand glaciers at the ACR (ref. 28), and while we cannot yet offer the precision of other studies, the geomorphic prominence of the two moraines supports a hypothesis for a broadly comparable two-stage lateglacial history (Fig. 8b). Importantly, core PS81/265-1 retains evidence for relatively proximal glacimarine sedimentation either side of ~10.7 cal ka B.P. which suggests that the fjords were still heavily influenced by glacimarine processes at this time. Thus the M1 moraine formation may date towards the younger part of the lateglacial, perhaps reflecting glacier response to increasing temperatures following ACR cooling, or pointing to a role of fjord geometry in stabilising recession. Whether a single or two-step event, the moraine chronology presented here extends the geographic footprint of the ACR into the Atlantic sector of the Southern Hemisphere for the first time (Fig. 1a)[29], and offers further direct confirmation of advance of glaciers at the mid-latitudes during this chronozone.

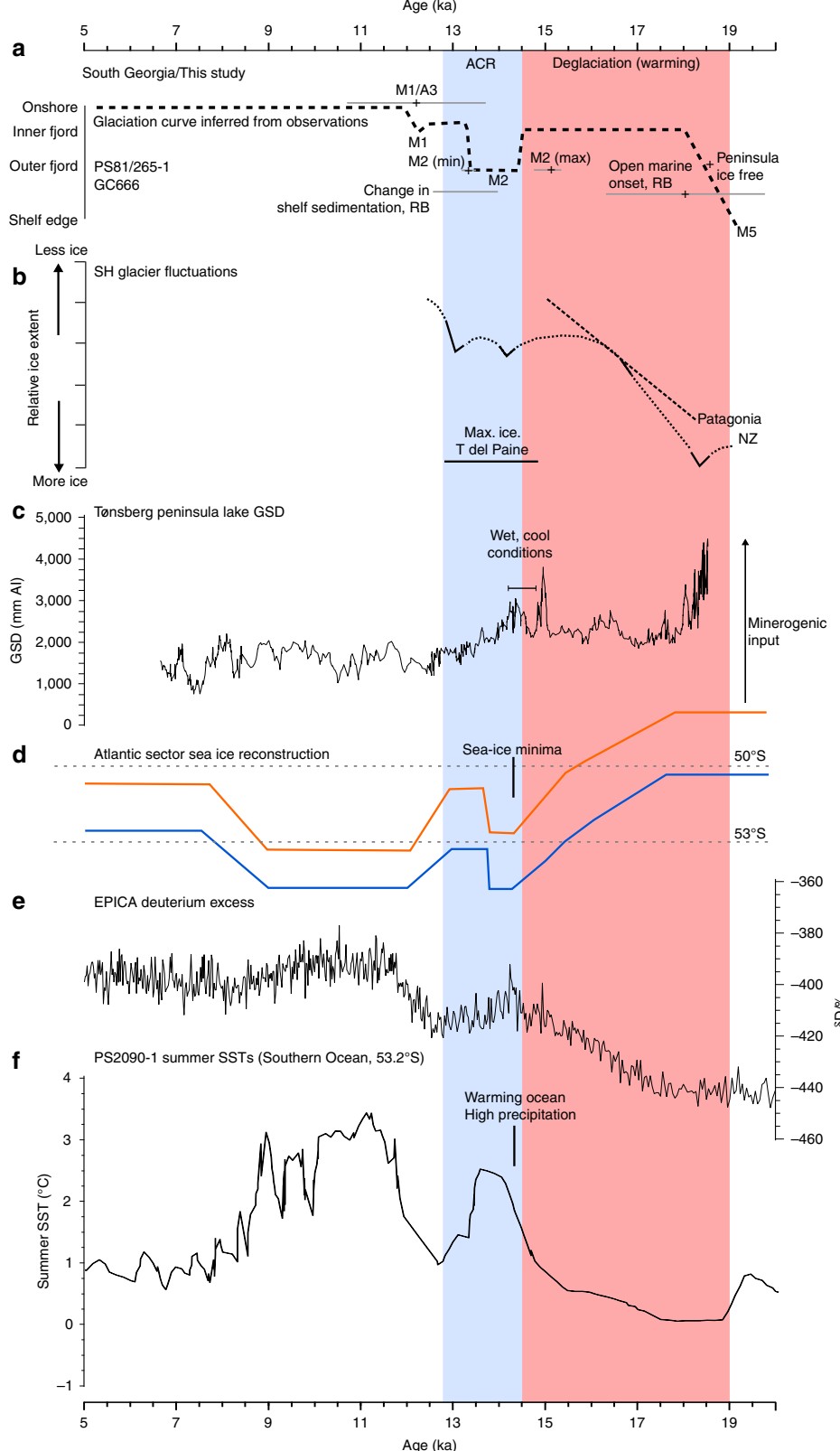

**Figure 8 | South Georgia glacial history compared against published Southern Hemisphere glacial and climatic records.** (**a**) Age constraints for events during the last deglaciation and termination on South Georgia from this study and published records, and a new glaciation curve inferred from these observations. These events are compared with (**b**) glacier fluctuations from other Southern Hemisphere sites (refs 28,34,73), (**c**) grey-scale density (GSD) record from Tønsberg Lake, north South Georgia, as a proxy for minerogenic input (ref. 25), (**d**) reconstructions of the northern sea-ice edge (blue line) and southern boundary of the polar front (orange line) for the last deglaciation in the Atlantic sector of the Southern Ocean (ref. 75), (**e**) EPICA Dome C (EDC) Ice core Deuterium content record, and (**f**) Summer sea-surface temperature (SST) record for core PS2090-1, south of the Antarctic Polar Front (ref. 75).

By comparison to other records of Southern Hemisphere glaciation, our proposed record of Late Quaternary glacier fluctuations appears similar to that of both New Zealand and Patagonia (Fig. 8). The ice cap that expanded across the South Georgia shelf at the LGM is comparable to the expansion of the northern Antarctic Peninsula Ice Sheet[70] and South Orkney Ice Cap[71] which were both significantly advanced up to ~19 cal ka B.P.[6,17]. Our estimates from fjord sediment infills seem to indicate that ice had already retreated to at least fjord mouth locations by ~18 cal ka B.P., consistent with lake records which indicate that adjacent peninsulas became ice free ~18.6 cal ka B.P.[25] (Fig. 8a). To reconcile these apparently early ice-free conditions requires a relatively early and rapid shelf deglaciation, alongside considerable thinning of the ice-surface profile in order to maintain fjord glaciers while leaving the present-day peninsulas deglaciated. We suggest that the ice cap that formed at the LGM which, based on the distribution of submarine landforms, had marine margins around its entirety, may not have been sustainable for long; the limited supply of moisture from the presence of winter sea ice[72], and widespread ablation by iceberg calving and ocean melting would have required sustained nourishment to maintain balance. Early and moderately rapid retreat after the LGM may have been forced by a combination of early deglacial warming at lower latitudes and elevations, the 'low-slung' profile which we have inferred for the outer margins of the ice cap, the sensitivity of the ice-cap to external forcing, and limits on precipitation supply, which is likely to have been transient and shifted with the latitudinal position of the Southern Westerlies.

Boex et al. recently showed that the rapid thinning and retreat of the Patagonian Ice Sheet initiated at 18.1 ka and reached near-present limits by 15.5 ka (ref. 73) (Fig. 8b). The mechanism to explain this rapid demise was warming associated with a southward isotherm shift, coupled with the poleward migration of the Southern Westerlies which had sustained the LGM ice field. Given that initial retreat around South Georgia was apparently slightly earlier than at more northern latitudes, deglaciation may not have been forced atmospherically but instead ocean-led driven by warming seas or rising sea-levels. Meltwater pulses sourced from Antarctica are one plausible trigger for initiating landward ice-cap retreat, with a significant iceberg rafting event (AID8) dated at ~19 cal ka B.P. recorded in marine sediments from the nearby Scotia Sea[74].

By contrast, our records for the subsequent ACR appear remarkably in phase with those of Patagonian mountain glaciers. Glacier advance in Torres del Paine culminated at 14.2 ± 0.56 ka and deglaciation had occurred by 12.5 ka (ref. 33), consistent with our proposed chronology (Fig. 8). Conditions responsible for the early lateglacial expansion are thought to be atmospheric, linked to northern migration of the south westerly wind belt to the latitude of Torres del Paine at the onset of the ACR. At 51°S, the latitude is near coincident with South Georgia to the east. A stronger precipitation-bearing Southern Westerlies influence at this latitude is therefore likely for the ACR on South Georgia too. In marine sediment cores recovered from south of the APF, surface water temperatures were actually warmer during the ACR than in the period prior to it (earlier than ~15.5 ka B.P.; Fig. 8f)[75] while sea ice was at a minimum during this time period in the Southern Ocean (Fig. 8d)[75]. We consider that this unusual high precipitation, warm maritime ocean and sea-ice minimum configuration is likely to have pre-conditioned South Georgia glaciers to respond sensitively and rapidly to the temperature drop that accompanied the ACR. Frontal shifts and migration of cool Southern oceanic waters are alternative drivers for the glacier expansion but the thermal reversal in the ocean is thought to lag behind the ACR by ~800 years[76] and is therefore unlikely to

be a key driver based on our interpreted chronology of events. Sea ice re-growth and reduction in precipitation supply around ~13.5 ka, may be subsequent triggers for the initial post-ACR retreat prior to the temperature increases seen at the end of the chronozone (Fig. 8e).

Irrespective of the principal driver, in the context of Andean, New Zealand and South American observations of ACR glacier advance, our results from South Georgia provide further support to the notion that the cooling was broadly synchronous across the southern mid-latitudes and induced a similar response in Southern Hemisphere glacial systems, whether in high mountain regions, or in oceanic regimes at sea level as suggested by this study.

Finally, a short glacial maximum and observations of regions of moderately shallow sea bed retaining no expression of glacial disturbance (Fig. 3a) reconcile our new interpretations of an expansive LGM with evidence for the persistence of shelf-inhabiting taxa on South Georgia, including the giant sea spider Colossendeis megalonyx, the limpet Nacella concinna[77] and Southern Ocean shrimps[78], through multiple glaciations. Our findings also have implications for the resilience of terrestrial biota which appear to have survived through at least one significant cycle of ice-cap advance and retreat[8].

## Methods

**Multibeam and sub-bottom profiler data acquisition.** New multibeam swath bathymetry data were acquired on cruise ANT-XXIX/4 of RV *Polarstern* in 2013 using a hull-mounted Atlas Hydrographic Hydrosweep DS III. The system was operated in hard-beam mode with a frequency of 15.5 kHz, 320 depth points per ping (across the swath track) at 2 × 2° beam resolution, with a maximum swath width of ~100°, leading to a swath width a little over twice the water depth. Vertical resolution is better than ~1% of the water depth. Sound velocity profiles collected using a sound-velocity probe were used to calibrate the sensor data. Rail tracks running along the nadir in the data set in Fig. 5b result from problems with the instrument's bottom detection algorithms during acquisition. Rather than remove the centre tracks entirely (and in-doing-so compromise visual coverage), we leave the artefacts intact whilst acknowledging their presence.

Existing but unpublished multibeam data, often comprising of single swaths across the continental block, were also assembled from a number of past cruises and gridded together with new data to form one coherent dataset for the regional mapping of landforms. The majority of datasets comprised cruises of the RRS *James Clark Ross* (British Antarctic Survey) acquired from 2000–2012. We used the software *MB-system* for the processing and output of gridded data. Bathymetry data were gridded at resolutions appropriate for the water depth of the working area and the quality/generation of echo sounder used during acquisition: normally equating to grid cell sizes between ~20 and 5 m. The coverage of data is non-uniform across the shelf, with greater coverage in the west and northern areas, as well as localized data surveys to the south. Few data exist from the east of the continental shelf (Supplementary Fig. 1).

For regions lacking multibeam coverage, we made use of the *Olex* data set, which provides gridded singlebeam echo-sounder bathymetry data, collected by fisheries and commercial vessels, at nominal spatial resolutions up to ~5 m (ref. 79). Coverage with *Olex* is relatively good along the outer shelf, north of South Georgia, where a commercial fishery is active.

Sub-bottom profiling data were acquired additionally on RV *Polarstern* using the parametric hull-mounted ATLAS PARASOUND P70 system. Primary operating frequencies were 18.75 and 22.95 kHz with a secondary frequency of 4.2 kHz. Vertical resolution is better than 0.2 m. Data visualization and processing were carried out using SeNT software (developed by H. Keil, University of Bremen). The vertical scale on profiles has been converted from travel time to metres using a constant sound velocity of 1,500 m s$^{-1}$.

**Landform mapping.** Sea-floor landforms were digitized as shapefiles from the acoustic datasets described above in ArcGIS. Moraines and bedforms were mapped as populations of polylines, following the crests of geomorphic elements, typically at a 1:25000 scale. Bedform amplitudes and geometry were extracted and assigned to individually mapped landforms across the shelf. We interpreted a number of sub-types of subglacial streamlined bedform from the mapped population. These have been broadly characterized and distinguished based on the following criteria: (i) roches moutonnées: streamlined ridge, with a smooth ice-moulded stoss face and contrasting irregular lee (inferred as the result of plucking). The landforms are assumed to be formed within indurated sedimentary rock or bedrock and have low elongations relative to other streamlined subglacial landform types. (ii) drumlins: low relief, ovoid hill or ridge, with a distinctive larger blunt end and narrowing, tapered lee. Elongated to stubby in relief. (iii) Mega-scale glacial

lineations: highly elongated (>10:1) parallel ridge-grooves, sometimes with a more prominent higher-relief head (seed). These are relatively infrequent on the South Georgia shelf but are clearly sediment-formed features where mapped.

**Core logging and analyses.** Two gravity cores used in this study were collected in 2012 and 2013 in Royal Bay and Cumberland Bay, respectively. Multi-proxy analyses included investigation of lithology, physical properties, grain counts and size distribution, clay mineral assemblages, facies succession, and radiocarbon dating of marine macrofossils. GC666 was acquired on cruise JR257 (35°44.4888′W, 54°25.236′S). The core was subsequently logged on a GEOTEK multi-sensor core logger at BOSCORF (NOC Southampton, UK), split, visually described, X-rayed and subsampled for analyses on discrete samples. X-rays were conducted on an NTB EZ240 scanner. Core PS81/265-1 was collected on ANT-XXIX/4 (36°26.5902′W, 54°14.1702′S). The core was analysed with a GEOTEK core logger and split onboard. It was subsequently described and sub-sampled for further analyses. Shear strength was measured using a hand-vain at 10 cm intervals. Counts on grains >2mm per centimetre of core were conducted from the X-radiographs[80]. Grain size was determined by sieving and settle tube separation of silt and clay. The ≤2 µm sediment fraction was used to determine the relative contents of the clay minerals smectite, illite, chlorite and kaolinite using a diffractometer system (Rigaku MiniFlex with CoKα radiation (30 kV, 15 mA)) at the Institute for Geophysics and Geology (University of Leipzig). The clay mineral identification and quantification followed the standard X-ray diffraction methods described by Ehrmann et al.[59]. C-org contents were analysed using an ELTRA CS-2000 after removal of carbonate with HCl. Biogenic opal was determined using established methods[81], and calculated with 10% (weight) water within the opal.

**Radiocarbon geochronology.** AMS 14C dating on samples from GC666 was performed at the Ion Beam Laboratory, ETH Zurich, Switzerland, using a 'mini radiocarbon dating—accelerator mass spectrometry' system ('MICADAS–AMS') equipped with a gas ion source. This array allows dating of exceptionally small sample quantities of carbonate material (refs 82,83). AMS 14C dating on PS81/265-1 was carried out at BETA Analytic, in Miami, using an equivalent micro-AMS service. Dates were analysed on picked specimens of foraminifera, shell or shell fragments across the two cores (Table 1). The conventional AMS dates were calibrated to calendar ages using the Calib 7.01 program and the Marine13 calibration data set. A marine reservoir correction of 1108 ± 61 yr was applied to all ages (determined using the mean of two surface ages measured on core top sediments from core GC666), taking into account that Calib operates with a standard global reservoir correction of 400 years (ΔR = 708 ± 61 yr). We present calibrated ages as a range, and as median values (95.4% confidence, 2σ).

**Data availability.** Gridded multibeam swath bathymetry for Cumberland Bay Trough are available online via PANGAEA (doi https://doi.pangaea.de/10.1594/PANGAEA.870752). Other geophysical and geological data sets generated during and/or analysed during the current study are available from the corresponding author on reasonable request.

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

## Acknowledgements

A.G. was supported by grants NE/K000527/1 and NE/K000527/2 from the Natural Environment Research Council (NERC), and by the British Antarctic Survey 'Ice Sheets' programme. The latter also supported contributions from D.A.H. and C.D.H. Fieldwork was supported by the Deutsche Forschungsgemeinschaft (DFG) in the framework of the priority programme 'Antarctic Research with comparative investigations in Arctic ice areas' by grant BO 1049/19. G.K. was supported by the Alfred-Wegener-Institute, Helmholtz Centre PACES II (Polar Regions and Coasts in the changing Earth System) programme, and by the European Commission under the 7th Framework Programme through the Action IMCONet (FP7 IRSES, action no. 319718).

## Author contributions

A.G. conceived the project (with input from C.D.H., D.A.H. and G.K.), conducted fieldwork (on JR257 and PS81), recovered the cores, oversaw and carried out many of the analyses and wrote the first draft of the ms. G.K. led the collection of and carried out physical core analyses on PS81/265-1 and supported the analysis and dating of GC666. O.M. carried out the analysis and initial interpretations of GC666. L.W. performed the radiocarbon measurements on GC666. W.E. carried out clay mineral assemblage analyses on PS81/265-1. P.W., C.d.S.F. and M.R. assisted in the collection and processing of hydroacoustic data on PS81. D.W. supported the fieldwork and advised on glaciological interpretations. G.B. was the principal investigator on cruise PS81 that allowed access to Cumberland Bay. All authors contributed to writing of the manuscript, and to the discussion of data and implications of results.

## Additional information

**Competing interests:** The authors declare no competing financial interests.

**Publisher's note**: 

