## [Peer Review File · Nature Communications]

Reviewer #1 (Remarks to the Author):

What are the major claims of the paper?

The paper presents new marine geomorphological and sedimentary evidence of former glaciation offshore of South Georgia, resulting in a major step forward in our interpretation of this region. In particular, they show evidence for pan-shelf ice-sheet advance and retreat and for a major re-advance during the Antarctic Cold Reversal.

Are they novel and will they be of interest to others in the community and the wider field?

These results are novel and of high interest in glaciology and climatology. The new geomorphological and sedimentological data put strong constraints on our understanding of glaciation in this key sub-Antarctic region.

Data and methodology?

Their interpretations are generally sound, although I do have a few concerns that I feel need to be addressed in the paper. In particular, although they present compelling evidence of shelf glaciation and for an Antarctic Cold Reversal Re-advance in the north-eastern sector of the ice sheet, I worry about extrapolating these results around the rest of the shelf. Many of the other areas display very different geomorphological signatures (e.g. southern sector where lineations are not confined to troughs) and I think more discussion is needed on whether the whole shelf was glaciated or not, highlighting some of the limitations and data poor regions. Moreover, recent work has highlighted the asynchronicity of glacier response due to factor such as topography. Lots of the moraines on the inner shelf occur at 'natural' pinning points and I therefore think it is dangerous to extrapolate a single chronozone to moraines occupying similar physiographic positions that may also have formed during retreat. Again, it would be good if there was some discussion on the limitations of their dataset, and if the interpretations were toned down a bit.

Appropriate use of statistics and treatment of uncertainty?

There is always some uncertainty in these type of geomorphological and sedimentological studies. But, the evidence in the north-eastern sector is compelling and well dealt with. I would, however, like the authors to acknowledge some of the uncertainties in other areas of the shelf, where there is less evidence.

Conclusions: robustness, validity, reliability?

Their conclusions are convincing (although see minor points above), and although there needs to be further work, particularly to provide more shelf edge and pan-island ages on retreat, it still presents a major step forwards in our understanding.

On a more subjective note, do you feel that the paper will influence thinking in the field?

Yes, it suggests a major change in our interpretation of glaciation on South Georgia and is therefore likely to have a major impact on the field.

The manuscript is well written and I only have a few minor comments, which I list below. Overall, if they can address these points I recommend publication in Nature Communications.

Specific comments

P2L48: I'm not sure I get this link to models for the Laurentide Ice Sheet? It doesn't seem to hang together with the rest of the sentence about optimising models for the Antarctic ice sheets. Needs more explanation or just delete.

P2L54: Delete "in situ"

P2L77: I am not sure references 14 and 15 are entirely appropriate. What about the recent reconstruction by Clark et al., (2012) of the British-Irish Ice Sheet in QSR, the Hughes et al. (2015) DATED time-slice reconstruction of the European ice sheets and the Margold et al. (2015) ice stream reconstruction of the north American ice sheets in Earth-Science Reviews.

P2L88: The treatment of ages varies considerably throughout the manuscript, e.g. ka BP, 1 sigma range, yrs ago, cal. ka BP. Please try and be consistent to help the reader.

P3L125: The troughs are not mapped on Fig. 1c. Could they be included, perhaps with a dotted line along the trough sides?

P3L139: This needs some clarifying - surely ice switching of palaeo-ice flow into these secondary troughs is not likely to happen between glacial episodes?

P3-4L141-142: What about geological controls on trough formation?

P4L160: See comments below about defining roche moutonee, crag-and-tail and drumlins in Figure 2. In particular, it would be nice to show some acoustic data to show whether they are formed in bedrock or sediment.

There are a lot of streamlined landforms in the southern part of the study area that are not associated with troughs and where moraines are rare. Can you account for or comment on these different geomorphic signatures?

P5L193-194: What is the evidence for this? Do you have acoustic data that shows the moraines continuing under the sedimentary fill?

P5L212: Looks more like a sandy silt diamicton.

P5L213: "9 cm"

P5L223: What about debris flows rather than melt out?

P6L264: Could you make a back of an envelope calculation to check this by counting the number of layers and comparing with the sedimentation rates deduced from the ages?

P7L397: Such as? Need to present an example of a truncated moraine or reference a paper here.

Figure 1 – The red stars need defining in panel a (especially as one star seems to relate to a core and the others to ages). I also think the references are wrong in this panel – for instance, #6 is not referred to and the reference used for New Zealand seems to relate to an age in Chile! In panel c it is very difficult to see the core locations – can the symbol size be made larger? I also think it would be good to include a map showing the cruise lines and different data sources. The authors have put a lot of effort into collecting and compiling the data, but it is not immediately clear how extensive it is.

Figure 2 – I am struggling to tell the difference between drumlins, crag-and-tails and roche moutonne in these sub panels. What criteria allow you to differentiate these different features, and in particular, do you have acoustic data that allows you to see if these features are cored by sediment or bedrock?

Figure 4 – There appears to be a quite a sharp shift from SW-NE orientated ridges further seaward into NW-SE orientated ridges closer to the land. Could these represent different glacial periods?

Supplementary A. What do the different colour arrows mean? The figure is also missing a legend, e.g. of the bathymetry. The secondary NW-SE orientated trough are very prominent. Is there a geological control on these?

- Stephen Livingstone

Reviewer #2 (Remarks to the Author):

This paper provides strong geomorphic evidence for an advance of the South Georgia Ice Cap to the outer platform in the past. However, none of the glacial landforms on the outer platform (shown in Figure 1) were dated. The only two age constraints come from two sediment cores collected on the inner, more shallow part of the platform and only one of these cores sampled a contact between till/proximal glacimarine sediments and open marine sediments. The authors thus rely on the argument that there are no "clear moraine limits" seaward of the fjord mouths to support platform-wide ice cap expansion during the LGM. I fail to see where this is demonstrated; certainly not in figures 1 or 4. Hence, the evidence for an expansive ice cap grounded to the edge of the island platform during the LGM is not compelling.

The basis for arguing for a re-advance of the ice cap, exclusive of peninsulas, during the Cold Reversal is a bit more convincing, but less than compelling. The authors state that the presence of shell fragments in diamicton and lack of bioturbation in core PS81/265-1 is best explained as reworking from fjord sediments during ice re-advance as opposed to overall retreat-it could simply reflect minor oscillations of the grounding line at this one location.

The Parasound profile shown in Figure 1a shows a prominent angular unconformity at the contact between the two units sampled in core GC666, with strong reflections that appear to dip toward the NW that are deeply truncated and overlain by a horizontally bedded unit. The surface is manifest in the core by all of the proxies, including a change in shear strength and a peak in sand. This contact is interpreted, probably correctly, as a marine surface of erosion. This all seems quite logical, but the rather convoluted argument that this surface somehow relates to the ACR is not convincing.

In summary, this paper presents some spectacular, and to my knowledge previously unpublished, multibeam images of glacial features that extend to the outer edge of the South Georgia platform. However, the age of these features remains, in my opinion, problematic. The argument for significant re-advance during the ACR is also weak. Given these limitations, I can not recommend publication in Nature.

Reviewer #3 (Remarks to the Author):

Review: Major advance of South Georgia glaciers during the Antarctic Cold Reversal following extensive sub-Antarctic glaciation.

Alastair Graham et al.

General impression:

This is a very well-written and well-presented manuscript, which can claim to decisively answer an outstanding question concerning the response of ice masses on South Georgia to the last global glacial period and the warming that followed. It completely revises the prevailing model of limited glaciation in favour of a major expansion, and it finds a coherent glacier response to the ACR signal. The climate responses of sub-Antarctic areas are of major importance in assessing how far the Southern Hemisphere/Antarctic climate 'mode' extends northward, in assessing the magnitude and drivers of an Antarctic mode of climate forcing, and in assessing the magnitude and drivers of a so-called bipolar seesaw. The findings of this paper are therefore of importance beyond its immediate (glacial geology) field. The conclusions are a significant revision of thinking locally, and will influence future scientific thinking about regional and global climate behaviour.

This article builds on recent work by some of the same authors accessing the marine record of glaciation off South Georgia for the first time. As a body of recent and future work, it is exciting

and serves to answer major questions that can otherwise only be speculated upon while limited to terrestrial evidence. This paper represents a major contribution by evaluating *all* presently available multibeam bathymetry data from the S.G. continental shelf. The interpretations of the geomorphology are convincing and important in their own right. The most significant conclusions here address timing, and this rests on two cores. The sedimentological interpretations are well-supported by core analyses and the interpretations are convincing. The cores provide vital chronological constraints that are internally logical, are precision samples based on carbonate rather than bulk sediment, and I see no immediate reason to question their validity. A weakness, however, is that there are only two cores, and very little subbottom profiler data to spatially extend their stratigraphic information. The model therefore rests on only two sets of dates: one extrapolated 'date' for first retreat from the LGM, and just one maximum and one minimum date for the ACR. I wonder why only two cores have been collected/analysed/dated? And why so little subbottom data is presented, which I would have imagined would strengthen their arguments.

I'm a little surprised there's no discussion of outer shelf palaeo-flow structure, based on the bedform evidence that appears to be widespread. I appreciate this turns the focus away from the chronological aspects so perhaps is best addressed in a manuscript with a different focus, but it could help to support interpretations of a low ice surface profile and vulnerability to rapid retreat, which are invoked in this manuscript to explain chronological observations and the early/rapid deglaciation. The mapping distribution appears to cross banks as well as troughs - how widespread are ice streams vs slow flow, for example?

The following detailed comments are largely suggestions and queries to improve the clarity of the text and the arguments presented, rather than criticism of any substance. I think the interpretations are, for the most part, robust, the findings exciting and the discussion of wide interest.

In more detail, by line number:

Title: "Major advance" - I agree there is good evidence for a coherent and widespread margin associated with the ACR, and that it is an advance feature, but very little support is given for the amount of retreat prior to readvance, which would constrain the magnitude of the event and support statements such as "major" or "significant" advance. I have suggested one way this could be strengthened at the relevant point of the Discussion (lines 346-347), but this should be made clear in order to support statements relating to magnitude of the event.

30: clarification: "Glacial troughs began to infill with [what type of sediments? Postglacial, marine...]"

37-39: I think this final sentence in fact undersells what is really a wholesale reversal of the South Georgia limited LGM model.

60: "where future Antarctic environmental change would be first detected." Is this something you could come back to later? - We might expect, on this basis, S.G. to respond early to post-LGM warming. How should we expect it to respond to ACR cooling &/or post-ACR warming, relative to other sites, either Antarctic or sub-Antarctic? - And does it obey these expectations?

72-73: stress why this is a regionally-important sector of the Southern Ocean - the relationships between Atlantic and Antarctic climate 'modes' are imperative to understand to get a better grasp of Atlantic/Antarctic climate drivers.

75: "number, timing and extent of past glaciations"

99-102: again, I think you could raise an even broader importance here - that glacier responses are a manifestation of an Antarctic climate 'mode', that has implications for the processes that

govern southern hemisphere climate patterns and inter-hemisphere climate forcing

127 and throughout: a question of terminology, which doesn't need discussion in the paper (would largely distract from the agenda of the paper) but I wonder if the authors can justify: even at its most extended (interpreted to shelf-edge), the South Georgia ice mass is described as an ice cap rather than small ice sheet. To some degree this is just terminology, but on the other hand there are (conceptual?) implications for e.g. ice surface profile, flow structures such as ice streams.

130: how are these moraine banks defined, morphologically? Or acoustically?

139: delete "most likely" - there's plenty of evidence for flow switching within a glaciation

154-156: I think these sentences should be turned around, as you make several assumptions here (which may well be justified, but are not motivated in the text as it stands). E.g. One would expect to find streamlined bedforms, but to account for their absence it is necessary to appeal to post/non-glacial sedimentary infill. Remove "Holocene and pre-Holocene" - this is rather meaningless; simply state non-glacial.

157: "little to no sediment cover" - to me this suggests bedforms are bedrock moulded, or sediment sitting on a rock surface. Are you referring to the amount of till, from which bedforms are formed (several examples in Fig 3 appear to be sedimentary rather than bedrock - the presence of iceberg scours supports that), or post-glacial sedimentation that otherwise you assume has buried bedforms?

193, and Fig 4: I'm not convinced by the interpretation of a single receding margin landward of M3 -4-5 retreat ridges behind M3 are aligned with the larger moraine, across swath, but then there is a patch (where you interpret a change in orientation, hinged slightly in the west) that I wonder whether moraines actually cross-cut/overprint. This would point to a (local?) retreat-reorientation of flow-readvance sequence, rather than monotonic retreat.

206: Suggest a re-wording of the sentence opening to "A comparably positioned [and sized] moraines is known to exist..." This sentence is a critical point in the conclusions of this ms: that there is an 'equivalent' moraine correlated to the ACR around numerous S.G. fjords, indicating a whole ice cap/sheet response. The reference given clearly supports the statement, but I wonder if you could also show some further data (multibeam images) in this paper (Supplementary figure, if necessary) that illustrate this widespread feature.

221: missing word: "typical for a grounding-line..."

222: what do you mean "open glacimarine"? - specifically open water sedimentation, or a more generic distal glacimarine?

244: delete "k." (k. cal. ka.)

247 to end of paragraph: what would cause the initiation of marine sedimentation on top of the moraine at 2.2ka? Appealing to a hiatus must also explain why that hiatus should end. Could you reinterpret as a long period of ice shelf or permanent sea ice cover, which broke up at this time? Although I agree you would still need to invoke a very low sedimentation rate up until then. Hodgson et al 2014 presented TOPAS data from the inner parts of this trough. Does it extend to M2? Does the subbottom stratigraphy inward of M2 inform the core chronology interpretations in any way?

262 to end of paragraph: I find the core unit description rather confusing here - I suggest revisiting and rewording. E.g. the sentence beginning line 262 introduces 2 units, but I think only one is described here, or does the "alternating with" refer to Unit 2?

Paragraph beginning line 272: can you clarify the meaning of "productive" throughout this paragraph? - I'm not entirely sure whether you (always) refer to biogenic productivity, or are using the term generically in terms of production of any/all sediment by the glacial system. See also line 380.

278 and Figure 6: this looks to me strongly like an erosional unconformity, and indeed you refer to it as such later in the Discussion (371-375). This isn't just a change in sediment supply, but needs an agent to remove material. You discuss this later, but the explanation given in this paragraph (280-281) is rather vague - can you strengthen this?

301-302: See above re. "continuity of forms" - though I don't think that, if there is a small zone of overprinting landward of M3, this undermines your overall conclusion here, since we would expect a larger outer moraine of any major interval of advance (cf M2, or the outer shelf moraines). But I'm not sure that the geomorphology necessarily supports a single phase of deglaciation in itself - your other data are required to support this. Would there be a different geomorphological signature if the outer moraines were the retreat from, say, MIS6 and the fjord landforms (M2 & M1) were LGM and subsequent? There is a clear macro-topographical change between the outer trough/bank system and the inner fjords - is this an 'artefact' of a change in geology, or a product of real long-term fjord / outer shelf trough development? The latter would suggest a different occupation history and/or mode of ice flow/erosion - consistent with some glacial intervals resembling your ACR stage reconstruction, rather than full continental shelf glaciation? I suggest, finally, rewording "fresh signature", which is rather ambiguous.

312-313: This sentence is misleading - since you refer to 'last termination', I mistook your reference to basal ages for those you derive for the base of the sediment column, based on sedimentation rates. Suggest you clarify you refer to core basal ages, and remove reference to 'last termination'. The period of time you have data for is much more limited than this.

332: typo Isotope

335: be consistent in which chronological values you use - younger than 15.2 cal ka.

346: "The maximum age for moraine M2...".

347: At this stage, you have not constrained the magnitude of any ACR readvance, so I don't think it is justified to refer to a "significant advance". For this some picture of retreat prior to the readvance is required. Here, I suggest bringing in paragraph lines 395-401, which contribute to this question and which sit rather out of place where they are currently positioned later in the Discussion. What is the distance that these truncated moraines would imply for the retreat-readvance sequence? Would you consider this a "significant" advance?

358-360: the relevance here is that the site was not overrun during the ACR, not that it shows an early retreat from the LGM.

363-366: this provides an interesting context, but I think is a little disingenuous here, as the ice cap did not (likely) readvance from a present-day configuration - or at least, you have not shown that it would have attained a present-day configuration during the first retreat. Can you constrain the magnitude of change relative to a possible pre-ACR configuration?

It would be out of place in this section, but could well be considered elsewhere: I wonder what the equivalent figures are for loss of ice from the LGM, especially given that post-LGM retreat is early? - Based on the (conservative) reconstruction of ACR limits in Supplm Fig B, and the position of outer shelf moraines, it appears that a massive amount of ice must be lost early in the post-LGM period - can you put any numbers to that?

374: what is the chronological evidence for a hiatus, that would appear any differently if there had been an erosional event? You go on here to invoke bottom current activity, which "appears not to have removed a significant amount of material". I agree with this latter interpretation - the acoustic stratigraphy in Fig 6 looks strongly like there is an erosional unconformity, though your ages indeed point to only a small removal of time. But I don't think you have explicit chronological support for a hiatus.

383: "remain low in GC666"

384: here and elsewhere - can you either define the way in which you use "lateglacial", or avoid the term. It has specific chronological significance to some, but I think it's being used in a more generic way here.

379 and onwards: your interpretation sounds entirely plausible. How does this relate, however, to core PS81/265-1, from the neighbouring major trough and which requires very low sedimentation rates through deglaciation and much of the Holocene? If you invoke oceanographic/climate conditions to account for the stratigraphy and sedimentation at GC666, to what extent should those be mirrored at PS81/265-1? Is there anything in the setting or the glacial environment that should account for their differences?

407: It is the terrestrial moraines that have been dated, not M1.

425: requires early and/or rapid deglaciation and with considerable thinning of the ice profile, to allow large fjord outlets which yet leave the present-day peninsulas ice-free. The 'low slung profile' is surely the only way to be able to reconcile the terrestrial lake chronology?

441: is there any offshore core evidence for warming oceans at this time? Steady back-stepping and creation of moraines suggests sea level and its relation to ice thickness weren't abrupt drivers of retreat, otherwise we might expect to see a major step back of a calving margin (if there had been rapid lift-off), rather than the regular retreat with sufficient time and grounding pressure to create the regular sequence of moraines.

Methods: geophysics details for JR257?

Figure 2: these (d) look more like crag and tails or even streamlined bedrock than drumlins.

Supplm Figure A: what is the significance of the small coloured arrowheads?

Sarah Greenwood

Reviewer #1:

Main comments (3)

1. *Many of the other areas display very different geomorphological signatures (e.g. southern sector where lineations are not confined to troughs) and I think more discussion is needed on whether the whole shelf was glaciated or not, highlighting some of the limitations and data poor regions.*

We have added further description to the results (P4L168) section to outline the fact that some streamlined bedforms are not constricted to troughs, and use this furthermore in support of our 'low-slung' profiles interpretation (suggested by Reviewer 3).

With regard to more discussion of data poor regions, we have tried to focus in on one specific area of the shelf where we are certain the outer shelf story is well developed, and have tried to avoid going into too much discussion about areas of the shelf we have less information about. The reason we didn't present a full reconstruction for the island was to deliberately avoid over-interpreting our data. For those reasons, we wish to maintain a narrow focus on what we can say.

We have added further zones of 'no data' to the map in Figure 1c to try to be clear about data coverage, and we have also discussed these limitations briefly in the Methods section. We also added a new coverage map as a supplementary figure.

2. *dangerous to extrapolate a single chronozone to moraines occupying similar physiographic positions that may also have formed during retreat. Again, it would be good if there was some discussion on the limitations of their dataset, and if the interpretations were toned down a bit.*

Acknowledged. We added text to emphasise the striking similarity of these features in terms of their geometry and gross geomorphology, and also cite and follow terrestrial examples of where moraines in similar physiographic positions have been suggested to be correlative in time. We added another supplementary figure to highlight the fjord-mouth moraines (as requested by Reviewer 3).

3. *Like the authors to acknowledge some of the uncertainties in other areas of the shelf, where there is less evidence.*

We added text discussing some of our other uncertainties about the amount of retreat prior to the inferred ACR advance.

Minor comments and changes by line number:

P2L48: I'm not sure I get this link to models for the Laurentide Ice Sheet? It doesn't seem to hang together with the rest of the sentence about optimising models for the Antarctic ice sheets. Needs more explanation or just delete.
Deleted as requested.

P2L54: Delete "in situ"
Done

P2L77: I am not sure references 14 and 15 are entirely appropriate. What about the recent reconstruction by Clark et al., (2012) of the British-Irish Ice Sheet in QSR, the Hughes et al. (2015) DATED time-slice reconstruction of the European ice sheets and the Margold et al. (2015) ice stream reconstruction of the north American ice sheets in Earth-Science Reviews.

We added two of these suggested references, removed ref 15, but kept ref 14 because we believe it shows how models and data have been integrated already in these locations to develop a much better understanding of ice-sheet dynamics. This is the type of work which demonstrates how far behind we are at South Georgia.

P2L88: The treatment of ages varies considerably throughout the manuscript, e.g. ka BP, 1 sigma range, yrs ago, cal. ka BP. Please try and be consistent to help the reader.

We have worked through the manuscript in order to be more consistent with our reporting of ages. We note that some of these variants are required – e.g. cosmogenic ages are treated 'ka' rather than 'ka BP', and where we are discussing a time period instead of reporting an age.

P3L125: The troughs are not mapped on Fig. 1c. Could they be included, perhaps with a dotted line along the trough sides?

We attempted this revision but the dashed lines covered up some of the moraine mapping and detracted from the more subtle features we wished to highlight. The colour-shading in the bathymetry and contour overlay already does a good job of picking out the locations of the troughs so we have left the figure as it was.

P3L139: This needs some clarifying - surely ice switching of palaeo-ice flow into these secondary troughs is not likely to happen between glacial episodes?

Dowdeswell *et al.* (2006) showed how ice streams can switch flow paths between glacial periods because of variations in the patterns of deposition (and erosion). It seems inconceivable to us that two troughs can be eroded out and developed to such an extent within the space of one glacial cycle. We tried to leave both options open in the text.

P3-4L141-142: What about geological controls on trough formation?

We showed in a previous paper that the primary troughs are indeed fault controlled (Graham *et al.* 2008) and more detailed discussion on the evolution of the shelf will make some fascinating future work. Here, I don't think the controls are really the focus, and additional discussion risks diverting our attention away from the main message: simply, the complexity of the shelf morphology. We don't have any constraint on the outer shelf sub-surface geology (from seismic for instance) so any further discussion is likely to veer into speculation.

P4L160: See comments below about defining roche moutonnee, crag-and-tail and drumlins in Figure 2. In particular, it would be nice to show some acoustic data to show whether they are formed in bedrock or sediment.

Agreed. We have added text to the landform mapping section of the methods in order to make some clearer distinction of the various bedform types. Again, too much focus on the interpretation of different bedform types in the main text would risk us losing focus on the main aims of the paper, so we opted to include this outside the flow of the main manuscript sections.

Unfortunately we don't have sub-bottom data for the areas of subglacial bedforms mapped in this study (the opportunistic nature of data collection meant most of the surveys were carried out with other objectives in mind; e.g. biological survey).

There are a lot of streamlined landforms in the southern part of the study area that are not associated with troughs and where moraines are rare. Can you account for or comment on these different geomorphic signatures?

We commented on the southern bedform features in the revision, and have used them to further support interpretations of the distribution of fast-flow and low-slung ice surface profiles. The lack of moraines may be due to poorer data coverage rather than differences in ice dynamics. Until we get more data, I'm reluctant to speculate too much on whether there really is a north-south difference.

P5L193-194: What is the evidence for this? Do you have acoustic data that shows the moraines continuing under the sedimentary fill?

We presume this to be the case. We do not have acoustic data to verify this at the moment. In the trimming process, we have actually removed this sentence anyway.

P5L212: Looks more like a sandy silt diamicton.

The matrix is certainly muddy (silt and clay), I agree, but I would prefer to avoid over-describing the diamicton - it could even be referred to as a sandy silty clayey diamicton! I clarified that the matrix is muddy in the line below this one, and retained the sandy diamicton descriptor.

P5L213: "9 cm"

Changed.

P5L223: What about debris flows rather than melt out?

A fair suggestion, and we offer that as an alternative too in the revised text.

P6L264: Could you make a back of an envelope calculation to check this by counting the number of layers and comparing with the sedimentation rates deduced from the ages?

This suggestion was already based on an assessment of the sedimentation rates derived from radiocarbon dating which indicates that annual layers may be likely and seasonal layers plausible too. I don't see much value, at least in this paper, of taking the discussion any further.

P7L397: Such as? Need to present an example of a truncated moraine or reference a paper here.

We already reference Hodgson *et al.* (2014) in which we showed many examples of truncated moraines. We also refer directly to the position of truncated features in Cumberland Bay in our revision.

Figure 1 – The red stars need defining in panel a (especially as one star seems to relate to a core and the others to ages). I also think the references are wrong in this panel – for instance, #6 is not referred to and the reference used for New Zealand seems to relate to an age in Chile! In panel c it is very difficult to see the core locations – can the symbol size be made larger? I also think it would be good to include a map showing the cruise lines and different data sources. The authors have put a lot of effort into collecting and compiling the data, but it is not immediately clear how extensive it is. Defined stars in caption and changed core symbol to circle. Checked and amended references. Increased size of core symbols on Figure 1c. We added a new supplementary figure during the revision (and reordered the existing ones) to show multibeam and Olex data coverage on the continental shelf. This addition ought to go some way to also addressing one of the main comments of Reviewer 1 – to highlight regions that are data-poor.

Figure 2 – I am struggling to tell the difference between drumlins, crag-and-tails and roche moutonne in these sub panels. What criteria allow you to differentiate these different features, and in particular, do you have acoustic data that allows you to see if these features are cored by sediment or bedrock?

Answered above in relation to the comment on P4L160. Further explanation provided in the Methods section.

Figure 4 – There appears to be a quite a sharp shift from SW-NE orientated ridges further seaward into NW-SE orientated ridges closer to the land. Could these represent different glacial periods?

We interpret this pattern to result from minor flow reconfiguration during overall retreat from the LGM, as the ice cap receded out of the secondary trough into the primary. We would like mention this explicitly in the text but have no room to do so. It is clear from the wider mapping work in Fig 1c and Fig 4a that many of the outer shelf ridges in the secondary trough wrap around towards a NW-SE direction as they mantle the large moraine banks, so we'd expect the component of NW-SE ridges to become more dominant as the ice margin approached the primary trough. I don't think we have any evidence that the landforms relate to different periods of glaciation.

Supplementary A. What do the different colour arrows mean? The figure is also missing a legend, e.g. of the bathymetry. The secondary NW-SE orientated trough are very prominent. Is there a geological control on these?

We added a bathymetry key to supplementary Figure B (old Fig A) and explained the meaning of the arrows in the associated caption. See comments above in relation to geological controls (it's possible, but I don't think we have any information on the sub-surface structure that can help us).

Reviewer #2 (Remarks to the Author – 4 clear points raised):

Reviewer acknowledged that the paper presents some spectacular, and previously unpublished datasets. Some negative comments but only a few clear suggestions for revision. We respond as best we can to the criticisms and have considered these in our revision.

4. none of the glacial landforms on the outer platform (shown in Figure 1) were dated.

This is true, and we have made no secret of the lack of dating of the outer shelf features in the manuscript. We attempted to core and sample landforms on the outer shelf during two previous expeditions, but have not had success to-date. Thus, we have tried to use a combination of direct age constraints from fjord sediment cores, as well as intelligent use of sedimentation rates applied to acoustic profiles, to provide a chronology as well as independent tests of that chronology. We have acknowledged that, as with many studies of this kind, there are still some limitations with the age data, but we offer a robust interpretation of the data that we have collected. Furthermore, we have attempted to improve our age constraints further still during the revision process by obtaining additional ¹⁴C age measurements. This, in our eyes, strengthens to case substantially for ACR advance as well as the evidence for a more extensive shelf-wide glaciation.

5. The authors rely on the argument that there are no "clear moraine limits" seaward of the fjord mouths to support platform-wide ice cap expansion during the LGM. I fail to see where this is demonstrated

We are unclear of the reviewers' argument here; there seems to be some confusion. We have stated in the text that there are in fact CLEAR moraine limits seaward of the fjord mouths. Since we are able to date the fjord mouth moraines to a post-LGM interval of time, we infer (quite reasonably) that the LGM must relate to a more extensive ice-cap limit seaward of the fjord regions. We make a reasoned argument why this must correspond to one of the outer platform moraine ridges that we have mapped. We note that the other two reviewers found our arguments convincing.

6. The authors state that the presence of shell fragments in diamicton and lack of bioturbation in core PS81/265-1 is best explained as reworking from fjord sediments during ice re-advance as apposed to overall retreat-it could simply reflect minor oscillations of the grounding line at this one location.

We have now added to the chronology of core PS81/265-1, specifically dating the transitional unit above the diamicton, thus improving our constraints on the ages of the various core units.

We made a clear argument that shell material found within this sediment deposit is most likely to be reworked from a pre-existing deposit. We see few other mechanisms for incorporating shells into till. The reasoning for an ACR advance rather than a stillstand has been built upon in our revision, and is supported by a number of observations, including the presence of truncated moraines in the fjord deep basins. Nevertheless, we have also included a statement in our revised manuscript that we cannot be certain about the nature of any retreat prior to the ACR, and thus leave open the possibility that the moraine relates to a more minor readvance instead (even though we argue otherwise).

7. *The Parasound profile shown in Figure 1a shows a prominent angular unconformity at the contact between the two units sampled in core GC666, with strong reflections that appear to dip toward the NW that are deeply truncated and overlain by a horizontally bedded unit. The surface is manifest in the core by all of the proxies, including a change in shear strength and a peak in sand. This contact is interpreted, probably correctly, as a marine surface of erosion. This all seems quite logical, but the rather convoluted argument that this surface somehow relates to the ACR is not convincing.*

It seems quite plausible to us that the change in sedimentation in GC666 relates to ice-cap proximity, and the timing of events fits with an ACR advance and retreat as we have laid out in the paper. (N.B. sedimentation rates also change through this interval, as well as the proxies). Indeed, we note that the reviewer agrees with our broad interpretation of the horizon. In the absence of a better or simpler suggestion for why there was a change in the shelf depositional regime at this time, we are happy and confident to maintain our original line of reasoning. The other two reviewers found our arguments convincing related to the ACR.

Reviewer #3

Main comments (2):

8. *A weakness is that there are only two cores, and very little subbottom profiler data to spatially extend their stratigraphic information. The model therefore rests on only two sets of dates: one extrapolated 'date' for first retreat from the LGM, and just one maximum and one minimum date for the ACR. I wonder why only two cores have been collected/analysed/dated? And why so little subbottom data is presented, which I would have imagined would strengthen their arguments.*

We have obtained 3 new radiocarbon dates for PS81/265-1 to improve its chronology in response to the editorial and reviewer comments, and thus can constrain our age model even further. This substantially improves our constraints on the ACR advance and, crucially, gives us an age for retreat from the outer fjord moraine as well. This should add compelling evidence to our existing arguments and satisfy the comments above.

We would like to have shown more core results. However, the collection and analysis of sediment cores is non-trivial, and we have focused particularly on those cores which provide clear constraints on ice-cap behaviour. The two sequences we have shown and their associated datasets represent a significant amount of field, laboratory, and analytical effort. With more resources and future coring campaigns we would hope to be able to extend this work further. However, we emphasise that at the moment, these two cores represent our best and ONLY direct datasets relating to the glacial history of South Georgia from the marine environment. No other cored sequences recovered basal tills, and attempts to core glacial sediments seaward of the fjords have, at the time of writing, been unsuccessful.

The sub-bottom data do exist for other parts of the shelf but have a relatively limited penetration. They are very good for showing the infill of troughs (as presented in our Figure 6a) but rarely give any insight into the detail of the sub-surface beneath the trough infill, and certainly not the generations of till that would make for a clear stratigraphic story. Therefore, for clarity, we have only used the sub-bottom profiler datasets as site context and to demonstrate the broad acoustic character of the trough sequences.

9. *I'm a little surprised there's no discussion of outer shelf palaeo-flow structure, based on the bedform evidence that appears to be widespread. I appreciate this turns the focus away from the chronological aspects so perhaps is best addressed in a manuscript with a different focus, but it could help to support interpretations of*

a low ice surface profile and vulnerability to rapid retreat, which are invoked in this manuscript to explain chronological observations and the early/rapid deglaciation.

The reviewer makes a good point, and we have tried to add a little more discussion about the flow structure in the south as support for the low-profile ice surface, as suggested. We do agree that too much discussion of the palaeo-flow structure would defer attention away from an otherwise concise story, and we acknowledge that this might be better treated in more detail in a future manuscript or report. For space and clarity, we have not expanded on the palaeo-flow dynamics too much.

Minor comments and changes by line number:

Title: "Major advance" - I agree there is good evidence for a coherent and widespread margin associated with the ACR, and that it is an advance feature, but very little support is given for the amount of retreat prior to readvance, which would constrain the magnitude of the event and support statements such as "major" or "significant" advance. I have suggested one way this could be strengthened at the relevant point of the Discussion (lines 346-347), but this should be made clear in order to support statements relating to magnitude of the event.

We added text to the main manuscript (as suggested later on by the reviewer) which constrains the magnitude of a likely readvance, and which supports us maintaining the word 'major' in the text and the title. We have also tried to bolster areas of the discussion and interpretation relating to the readvance, making it clear in the process that we do not have a good handle on the exact extent of initial retreat prior to the ACR advance.

30: clarification: "Glacial troughs began to infill with [what type of sediments? Postglacial, marine...]" Changed.

37-39: I think this final sentence in fact undersells what is really a wholesale reversal of the South Georgia limited LGM model.

Agreed. We were not keen to over-sell our findings in the original submission. We have now reworded the text in the abstract slightly to add emphasis to our extensive glaciation interpretation; in the discussion, we make it very clear that the results over-turn the vast majority of previous ice-cap reconstructions.

60: "where future Antarctic environmental change would be first detected." Is this something you could come back to later? - We might expect, on this basis, S.G. to respond early to post-LGM warming. How should we expect it to respond to ACR cooling &/or post-ACR warming, relative to other sites, either Antarctic or sub-Antarctic? - And does it obey these expectations?

Acknowledged. We have tried to extend discussion a little more, surrounding the ACR advance and retreat triggers.

72-73: stress why this is a regionally-important sector of the Southern Ocean - the relationships between Atlantic and Antarctic climate 'modes' are imperative to understand to get a better grasp of Atlantic/Antarctic climate drivers.

I think we probably emphasise the regional importance enough already without going too far into explaining the SH climate system.

75: "number, timing and extent of past glaciations".

I didn't change the text in this case, as we go straight on to state that extent is also unknown in the next few sentences.

99-102: again, I think you could raise an even broader importance here - that glacier responses are a manifestation of an Antarctic climate 'mode', that has implications for the processes that govern southern hemisphere climate patterns and inter-hemisphere climate forcing. I acknowledge the suggestion and it's a good idea. However, I'd like to avoid returning to 'wider importance' at this point in the text, given that the introduction has already narrowed to the ACR and its regional signal. If I go back to talking about inter-hemispheric connections, I feel I'd be going off-track.

127 and throughout: a question of terminology, which doesn't need discussion in the paper (would largely distract from the agenda of the paper) but I wonder if the authors can justify: even at its most extended (interpreted to shelf-edge), the South Georgia ice mass is described as an ice cap rather than small ice sheet. To some degree this is just terminology, but on the other hand there are (conceptual?) implications for e.g. ice surface profile, flow structures such as ice streams.

I agree. Many of the structures are indeed reminiscent of glaciated margins affected at the 'ice-sheet' scale. I've taken the definition of an ice cap in the literature (based purely on size: <30,000 km² in size) and applied it throughout. I checked the text to make sure this terminology is consistent.

130: how are these moraine banks defined, morphologically?

Yes, defined based on morphology, given the lack of acoustic penetration through them. We made this clear in the revised text.

139: delete "most likely" - there's plenty of evidence for flow switching within a glaciation.

Agreed. But I'm not aware of any evidence for flow-switching having carved out two substantial cross-shelf troughs during one single period of glaciation? The relevant part of the manuscript leaves both options open.

154-156: I think these sentences should be turned around, as you make several assumptions here (which may well be justified, but are not motivated in the text as it stands). E.g. One would expect to find streamlined bedforms, but to account for their absence it is necessary to appeal to post/non-glacial sedimentary infill. Remove "Holocene and pre-Holocene" - this is rather meaningless; simply state non-glacial.

Changed order of text as suggested, but kept the 'Holocene and pre-Holocene' statement as it is factually correct – we know there are Holocene sediments, but at this point in the text pre-Holocene also leaves open the possibility that much older sediments have covered over the landforms (including glacial stage deposits, even if these are also formed in a marine depositional environment).

157: "little to no sediment cover" - to me this suggests bedforms are bedrock moulded, or sediment sitting on a rock surface. Are you referring to the amount of till, from which bedforms are formed (several examples in Fig 3 appear to be sedimentary rather than bedrock - the presence of iceberg scours supports that), or post-glacial sedimentation that otherwise you assume has buried bedforms? We are referring to the latter, and have adjusted the text accordingly to add clarity.

193, and Fig 4: I'm not convinced by the interpretation of a single receding margin landward of M3 -4-5 retreat ridges behind M3 are aligned with the larger moraine, across swath, but then there is a patch (where you interpret a change in orientation, hinged slightly in the west) that I wonder whether moraines actually cross-cut/overprint. This would point to a (local?) retreat-reorientation of flow-readvance sequence, rather than monotonic retreat.

It is difficult to tell whether there is a slight overprint here, because of the data quality at the edges of the swaths (it draws the eye towards seeing some smaller ridges that are more artefact than real data). None of my mapped ridges are interpreted to cross cut. Nevertheless, I agree with the reviewer that there may be some local evidence for minor grounding line fluctuations during retreat, which result from flow reorganisation. However, the main argument is that these are interpreted to relate to a single broad phase of retreat – if this wasn't purely monotonic then our overall interpretation doesn't really change.

206: Suggest a re-wording of the sentence opening to "A comparably positioned [and sized] moraines is known to exist..." This sentence is a critical point in the conclusions of this ms: that there is an 'equivalent' moraine correlated to the ACR around numerous S.G. fjords, indicating a whole ice cap/sheet response. The reference given clearly supports the statement, but I wonder if you could also show some further data (multibeam images) in this paper (Supplementary figure, if necessary) that illustrate this widespread feature.

Changed the sentence structure and created a new supplementary figure as requested.

221: missing word: "typical for a grounding-line..."

Changed.

222: what do you mean "open glacimarine"? - specifically open water sedimentation, or a more generic distal glacimarine?

Changed to make this clear.

244: delete "k." (k. cal. ka.)

Done.

247 to end of paragraph: what would cause the initiation of marine sedimentation on top of the moraine at 2.2ka? Appealing to a hiatus must also explain why that hiatus should end. Could you reinterpret as a long period of ice shelf or permanent sea ice cover, which broke up at this time? Although I agree you would still need to invoke a very low sedimentation rate up until then. Hodgson et al 2014 presented TOPAS data from the inner parts of this trough. Does it extend to M2? Does the subbottom stratigraphy inward of M2 inform the core chronology interpretations in any way?

The explanation we give in the paper relates to sediment focussing within the trough which, we interpret, leads to a period of non-deposition on the moraine flank (up until ~2.2 k when sediments onlap at the core site). The TOPAS data were reviewed again during the revision and we are unable to make a stratigraphic tie to the core using those data. The profile presented in Fig 6a show how sediment units onlap onto underlying highs, and we assume that this

is the same process going on in Cumberland Bay. This will certainly be an interesting subject to return to in future work.

262 to end of paragraph: I find the core unit description rather confusing here - I suggest revisiting and rewording. E.g. the sentence beginning line 262 introduces 2 units, but I think only one is described here, or does the "alternating with" refer to Unit 2?

The units are described in sequence; Unit I then Unit II. The alternations the reviewer refers to are within the unit. We re-read this section, and re-worded slightly to avoid any ambiguity.

Paragraph beginning line 272: can you clarify the meaning of "productive" throughout this paragraph? - I'm not entirely sure whether you (always) refer to biogenic productivity, or are using the term generically in terms of production of any/all sediment by the glacial system. See also line 380.

We clarify in the revision that we mean biological productivity in the ocean.

278 and Figure 6: this looks to me strongly like an erosional unconformity, and indeed you refer to it as such later in the Discussion (371-375). This isn't just a change in sediment supply, but needs an agent to remove material. You discuss this later, but the explanation given in this paragraph (280-281) is rather vague - can you strengthen this?

We avoided discussing the unconformity here in the results. I think it would damage the structure of the manuscript to begin trying to explain the unconformity at this juncture. However, in the revised text I have made it clear in this section that the unconformity records an erosional event (two additions to the paragraph).

301-302: See above re. "continuity of forms" - though I don't think that, if there is a small zone of overprinting landward of M3, this undermines your overall conclusion here, since we would expect a larger outer moraine of any major interval of advance (cf M2, or the outer shelf moraines). But I'm not sure that the geomorphology necessarily supports a single phase of deglaciation in itself - your other data are required to support this. Would there be a different geomorphological signature if the outer moraines were the retreat from, say, MIS6 and the fjord landforms (M2 & M1) were LGM and subsequent? There is a clear macro-topographical change between the outer trough/bank system and the inner fjords - is this an 'artefact' of a change in geology, or a product of real long-term fjord / outer shelf trough development? The latter would suggest a different occupation history and/or mode of ice flow/erosion - consistent with some glacial intervals resembling your ACR stage reconstruction, rather than full continental shelf glaciation? I suggest, finally, rewording "fresh signature", which is rather ambiguous.

The back-stepping pattern of moraines, not just in Cumberland Bay, but on other parts of the shelf too seem to all indicate a single major phase of shelf deglaciation. If there are a few smaller readvances within it then that is okay, and does not undermine the overall conclusion as the reviewer acknowledges.

We agree that a retreat pattern from an older glaciation may look similar to a comparable phase of retreat during the LGM-deglaciation, but the lack of degradation of the ridges and their sea-floor exposure (they are neither buried nor heavily eroded) seems to point towards them being more recent features. I added to the text to try and strengthen this argument. We then go on to support this argument with other data.

Unfortunately, we have no data on the geology to test the idea that some parts of the macro-topography are influenced by geological structure. A nice idea for future work though.

We changed "fresh signature" as requested.

312-313: This sentence is misleading - since you refer to 'last termination', I mistook your reference to basal ages for those you derive for the base of the sediment column, based on sedimentation rates. Suggest you clarify you refer to core basal ages, and remove reference to 'last termination'. The period of time you have data for is much more limited than this.

Changed the sentence to read 'latter part of the last termination' and clarified that we mean core basal ages.

332: typo Isotope
Changed.

335: be consistent in which chronological values you use - younger than 15.2 cal ka.

Changed.

346: *"The maximum age for moraine M2..."*.

Changed.

347: *At this stage, you have not constrained the magnitude of any ACR readvance, so I don't think it is justified to refer to a "significant advance". For this some picture of retreat prior to the readvance is required. Here, I suggest bringing in paragraph lines 395-401, which contribute to this question and which sit rather out of place where they are currently positioned later in the Discussion. What is the distance that these truncated moraines would imply for the retreat-readvance sequence? Would you consider this a "significant" advance?*

We have been able to frame the magnitude of the ACR advance in a clearer way in the revision and thus maintain that it was a significant event. We also flipped the paragraph order as suggested which brings in discussion of the truncated moraines much earlier than in the previous version.

358-360: *the relevance here is that the site was not overrun during the ACR, not that it shows an early retreat from the LGM.*

Text has changed to reflect the important points.

363-366: *this provides an interesting context, but I think is a little disingenuous here, as the ice cap did not (likely) readvance from a present-day configuration - or at least, you have not shown that it would have attained a present-day configuration during the first retreat. Can you constrain the magnitude of change relative to a possible pre-ACR configuration?*

We now have a sense of the pre-ACR configuration but not a good enough handle to put these sorts of numbers on the size of the ice cap. Therefore, I have kept this section in but made it clear by rewording that we're not saying the ice cap advanced from a present-day configuration.

It would be out of place in this section, but could well be considered elsewhere: I wonder what the equivalent figures are for loss of ice from the LGM, especially given that post-LGM retreat is early? - Based on the (conservative) reconstruction of ACR limits in Supplm Fig B, and the position of outer shelf moraines, it appears that a massive amount of ice must be lost early in the post-LGM period - can you put any numbers to that?

It is feasible to talk about the spatial extent/areal amount of ice lost, but I feel that without palaeo-thickness data we are risking veering into speculation by trying to fix numbers to something we still don't know that much about. A modelling investigation would be the best approach to answer these questions, but that is beyond the scope of the present study.

374: *what is the chronological evidence for a hiatus, that would appear any differently if there had been an erosional event? You go on here to invoke bottom current activity, which "appears not to have removed a significant amount of material". I agree with this latter interpretation - the acoustic stratigraphy in Fig 6 looks strongly like there is an erosional unconformity, though your ages indeed point to only a small removal of time. But I don't think you have explicit chronological support for a hiatus.*

This is a wording/meaning issue and we agree entirely with the reviewer. We actually meant to state that there is an absence of evidence for a hiatus. The way it was phrased originally suggested we were arguing for a hiatus when in fact we were proposing the opposite. We rephrased this to make it clear.

384: *here and elsewhere - can you either define the way in which you use "lateglacial", or avoid the term. It has specific chronological significance to some, but I think it's being used in a more generic way here.*

Our terminology adheres to the time period commonly recognised as relating to the lateglacial – c. 15-11 kyr.

379 and onwards: *your interpretation sounds entirely plausible. How does this relate, however, to core PS81/265-1, from the neighbouring major trough and which requires very low sedimentation rates through deglaciation and much of the Holocene? If you invoke oceanographic/climate conditions to account for the stratigraphy and sedimentation at GC666, to what extent should those be mirrored at PS81/265-1? Is there anything in the setting or the glacial environment that should account for their differences?*

Because GC666 is open shelf and core PS81/265-1 is in the fjord, we'd probably expect to see different things going on in both localities. We interpret the lower half of core GC666 and the oceanographic changes half way up the core to be taking place at the same time as we propose the moraine till is being deposited at PS81/265-1 so there should be no record of the lateglacial phase at the moraine site. It is true that the sedimentation in Royal Bay Trough through the Holocene isn't mirrored in PS81/265-1 as far as we can see, but we interpret this as a result of shut-off of sediment supply to the moraine ridge rather than to different sedimentation rates. The high sedimentation rates in

PS81/265-1 from ~2k onwards reflects proximity to two major tidewater fjord glaciers, whereas GC666 is more distal to the Holocene ice limits and shows a dominant hemipelagic style of Holocene sedimentation. In summary, the cores and the timing/sequence of events we interpret are consistent with one another given their respective locations on the shelf.

407: It is the terrestrial moraines that have been dated, not M1.

Agreed and edited.

425: requires early and/or rapid deglaciation and with considerable thinning of the ice profile, to allow large fjord outlets which yet leave the present-day peninsulas ice-free. The 'low slung profile' is surely the only way to be able to reconcile the terrestrial lake chronology?

Agreed, and changed.

441: is there any offshore core evidence for warming oceans at this time? Steady back-stepping and creation of moraines suggests sea level and its relation to ice thickness weren't abrupt drivers of retreat, otherwise we might expect to see a major step back of a calving margin (if there had been rapid lift-off), rather than the regular retreat with sufficient time and grounding pressure to create the regular sequence of moraines.

A good suggestion. We clarified in our revised text that perhaps sea level was the initial trigger, rather than the subsequent forcing factor. I don't think we are in a position to be able to correlate retreat with ocean temperature – we just don't have a good enough constraint on the post-LGM retreat right now.

Methods: geophysics details for JR257?

We didn't deem these necessary since we didn't show bathymetry data specifically from this cruise.

Figure 2: these (d) look more like crag and tails or even streamlined bedrock than drumlins.

Some discussion added during the revision to explain our differentiation of landforms.

Supplm Figure A: what is the significance of the small coloured arrowheads?

Now explained in figure caption.

Reviewers' comments:

Reviewer #1 (Remarks to the Author):

The authors have done a good job of responding to the reviewers comments and in revising the paper. In particular, they have responded well to the criticism about the lack of age control. This includes the addition of more dates and clarifying the importance of the geomorphological data.

Reading through I found a few small typos and also had a number of very minor edits that may prove useful (see below). But overall, I am happy for this to now be published in Nature Communications.

P3L126 - "...the primary trough."

P4L152 - needs a space before "Some"

P4L164 - "with corresponding low elevation lines of equilibrium" - I am not exactly sure what they mean here in terms of equilibrium. Could do with clarifying.

P7L319 - "our previously posed models" is vague and confusing being in the first person. Can you clarify as the "limited extent model"

P8L343 - delete "identical"

P8L345 - delete "induced"

P9L409 - space between "and" and "rapid"

P9L410 - space between "glacier" and "while"

P9L426 - space between "Antarctica" and "while"

P10L441 - two full-stops.

Reviewer #2 (Remarks to the Author):

Having read the revised version of the manuscript, I am satisfied that the authors have done an excellent job of addressing my concerns, and those of the other reviewers. The additional radiocarbon ages are quite significant in this regard and provide compelling evidence for the timing of the advance. I

I support publication of this paper.

John Anderson

Reviewer #3 (Remarks to the Author):

Review – revision of “Major advance of South Georgia glaciers during the Antarctic Cold Reversal following extensive sub-Antarctic glaciation”; Graham and others.

I am satisfied that the authors have made a considered response to the earlier set of reviews, and the additional radiocarbon dating in this revised version (should) strengthen(s) the authors’ interpretations by permitting a time for retreat from the apparent ACR advance position to be assigned. I have two further reservations about the chronology. I am convinced that the M2 position (and its corollaries) corresponds to a late-glacial ice margin. I also agree that the ACR offers a plausible time period for the interpreted advance. However, in strict and objective interpretation of the dates/stratigraphy:

- given that the moraine sediments (Unit I) contain reworked biogenic material, why should the ice-proximal forams in Unit II be in situ, since it is interpreted as comprising grounding line delivered sediments? If these were all reworked, the advance timing would be much later. Is there

any evidence from the condition of individual forams that point to the lower ones being reworked while the upper are in situ?

- what is the basis for interpreting minimal erosion of sediments across the unconformity at site GC666? The acoustics (Fig 6a) suggest that around 5m of ~horizontal strata are truncated by the overlying dipping unit. Does this undermine your interpreted chronology across this transition?

The manuscript is well-written, follows a logical argument and presents an important re-interpretation of sub-Antarctic glacial/climate history. I have only a few further, minor points to raise:

lines 129-132: this interpretation could be argued a bit more explicitly – I think it's a more powerful argument for multiple extensive glaciations than the cluster of outer shelf moraines (which could be fluctuations of a single, prolonged LGM). Trough dimensions (excavation of several 10s-100s m) can only be explained as long-term (multi-glacial) phenomena. Different glaciations could be expected to have had similar flow/retreat behaviour and comparable trough-to-trough dynamics, since secondary troughs are graded to a similar level, and display comparable landform assemblages from the most recent retreat

174-5: clearly several limits, but why not multiple overriding ridges formed during a prolonged but fluctuating MIS2?

195: "comprising three lithological units" (delete 'of')

224-5: strictly, the reworked shells simply state that the moraine relates to a margin position that occurred after ice-free conditions corresponding to the age of the shell, i.e. post ~15 ka. The chronology is what supports a readvance, rather than the reworking only. Old carbon could have been incorporated in a till prior to shelf-edge advance and still be deposited in the retreat moraines (e.g. the problem with bulk sediment dating approaches around Antarctica). The mere presence of the shells doesn't, strictly, mean that this particular moraine is an advance feature.

334-5: the age constrains an advance to *some time after* 15.2 ka, not, strictly speaking, the actual timing of advance. I think two things are important here in constraining an ACR event: i) given an LGM shelf-edge glaciation (well-argued), the shell dates around 15ka imply a period of ice retreat at this time, which must have been followed by advance, i.e. a readvance; and ii) given post-ACR dates in the unit above (assuming in situ – see comment above), these would *collectively* constrain the advance to ACR timing.

356: I think this is an important line of evidence in the model presented in this paper. While the paper referred to is authored by several of the same team, I would like to see key evidence illustrated in this manuscript, and not referred to another.

368: see main comment above – what is your basis for interpreting minimal erosion?

408: "To reconcile these..." (delete 'with'). And subsequent couple of sentences: there's some missing spaces (presume obscured by tracked changes) and a long and awkward sentence 410-4 – suggest reword this.

439: "warm maritime ocean and..."

Sarah Greenwood

Reviewer #1:

Typographic edits only (listed below). Reviewer satisfied otherwise.

P3L126 - "...the primary trough." We actually see no need for a second repetition of trough in this sentence.

P4L152 - needs a space before "Some". Changed.

P4L164 - "with corresponding low elevation lines of equilibrium" - I am not exactly sure what they mean here in terms of equilibrium. Could do with clarifying. We refer here to the equilibrium line of the ice cap's tidewater outlets. We thought this seemed clear on a second reading so haven't made any further edits.

P7L319 - "our previously posed models" is vague and confusing being in the first person. Can you clarify as the "limited extent model" Clarified in text.

P8L343 - delete "identical". Done.

P8L345 - delete "induced". Done.

P9L409 - space between "and" and "rapid". Done.

P9L410 - space between "glacier" and "while". Done.

P9L426 - space between "Antarctica" and "while". Done.

P10L441 - two full-stops. Changed.

Reviewer #2:

Reviewer satisfied with revisions. No changes.

Reviewer #3

Main comments (2):

given that the moraine sediments (Unit I) contain reworked biogenic material, why should the ice-proximal forams in Unit II be in situ, since it is interpreted as comprising grounding line delivered sediments? If these were all reworked, the advance timing would be much later. Is there any evidence from the condition of individual forams that point to the lower ones being reworked while the upper are in situ?

A sensible question. The sample we dated from the moraine till consisted of gastropod fragments that were broken and disturbed, whereas the sample of carbonate material from the overlying ice-proximal sediments consisted of intact and unbroken foraminifera. Hence, these are two very different types of carbonate material. In sediments recovered from similar ice-marginal settings (e.g. papers by Phil Bart on Ross Sea grounding zone wedges) well preserved forams in grounding-line proximal deposits are interpreted to have been emplaced at or near to the ice front. If the forams had been reworked subglacially, we would probably expect a poorer sample preservation, as we have observed for discrete samples within the moraine till that show more degraded and broken specimens. Forams in the upper ice-proximal facies are also more abundant than in the lower moraine faces, which is not consistent with their both being derived from a subglacial source. In summary, we are confident that the shell fragments in the till are reworked, whereas the forams that date in stratigraphic order in the overlying ice-proximal glacial facies reflect primary deposition in an open water glacier-proximal setting.

what is the basis for interpreting minimal erosion of sediments across the unconformity at site GC666? The acoustics (Fig 6a) suggest that around 5m of ~horizontal strata are truncated by the overlying dipping unit. Does this undermine your interpreted chronology across this transition?

The chronology of the core shows quite clearly that there has been minimal erosion at least in the immediate vicinity of the core location, as there is no hiatus (or if there is, then a very small time interval is missing) across the boundary. Hence, we can interpret the core record with confidence, which is the key argument for the current paper. On the profile we have shown some of the truncation is non-uniform, and there appear to be some sub-units that thin across trough rather than being removed entirely. There is certainly several metres of the lower unit missing in certain locations and future work will focus in greater detail on the spatial extent as well as significance of this horizon.

Minor comments and changes by line number:

lines 129-132: this interpretation could be argued a bit more explicitly – I think it's a more powerful argument for multiple extensive glaciations than the cluster of outer shelf moraines (which could be fluctuations of a single, prolonged LGM). Trough dimensions (excavation of several 10s-100s m) can only be explained as long-term (multi-glacial) phenomena. Different glaciations could be expected to have had similar flow/retreat behaviour and

comparable trough-to-trough dynamics, since secondary troughs are graded to a similar level, and display comparable landform assemblages from the most recent retreat

We added text to emphasise the interpretation as suggested.

174-5: clearly several limits, but why not multiple overriding ridges formed during a prolonged but fluctuating MIS2?

At this point in the text, the multiple ridges lend support to our previous interpretation of multiple glaciations, but the reviewer is right in that several could equally relate to a fluctuating MIS2. I don't think we have any evidence for that, so I'd prefer not to speculate and go with the simplest interpretation.

195: "comprising three lithological units" (delete 'of')

Done.

224-5: strictly, the reworked shells simply state that the moraine relates to a margin position that occurred after ice-free conditions corresponding to the age of the shell, i.e. post ~15 ka. The chronology is what supports a readvance, rather than the reworking only. Old carbon could have been incorporated in a till prior to shelf-edge advance and still be deposited in the retreat moraines (e.g. the problem with bulk sediment dating approaches around Antarctica). The mere presence of the shells doesn't, strictly, mean that this particular moraine is an advance feature.

We acknowledge the point here, and agree that in theory you could derive older carbonate material by reworking an existing till that was already deposited at the base of the fjord. In this case, however, I think our interpretation is consistent with both the facies (a muddy diamicton), the setting (fjord marine setting with high sediment flux) and chronology (as the reviewer points out) in suggesting reworking of a fjord glacimarine deposit.

*334-5: the age constrains an advance to *some time after* 15.2 ka, not, strictly speaking, the actual timing of advance. I think two things are important here in constraining an ACR event: i) given an LGM shelf-edge glaciation (well-argued), the shell dates around 15ka imply a period of ice retreat at this time, which must have been followed by advance, i.e. a readvance; and ii) given post-ACR dates in the unit above (assuming in situ – see comment above), these would *collectively* constrain the advance to ACR timing.*

Appreciated. I think we make it clear in various parts of the manuscript that the age of 15.2 ka is a maximum age (actually, we state this on line 334 and also show this in Fig 7), and that it constrains an upper age for the advance. We go on to constrain the lower limit in the following sections of text. In fact, I think we already outline the collective evidence which the reviewer is referring to in the comment. We just build up this picture over a number of paragraphs rather than in one short section.

356: I think this is an important line of evidence in the model presented in this paper. While the paper referred to is authored by several of the same team, I would like to see key evidence illustrated in this manuscript, and not referred to another.

Acknowledged, and therefore we added a further supplementary figure to show these promontories clearly.

368: see main comment above – what is your basis for interpreting minimal erosion?

See above.

408: "To reconcile these..." (delete 'with'). And subsequent couple of sentences: there's some missing spaces (presume obscured by tracked changes) and a long and awkward sentence 410-4 – suggest reword this.

Slight rewording to improve readability.

439: "warm maritime ocean and..."

Done.